# BOLTZMANN PRIORS FOR IMPLICIT TRANSFER OPERATORS

**Juan Viguera Diez**[1,2]    **Mathias Schreiner**[1]    **Ola Engkvist**[1,2]    **Simon Olsson**[1] [*]

[1] Department of Computer Science and Engineering Chalmers University of Technology
and University of Gothenburg SE-41296 Gothenburg, Sweden.
[2] Molecular AI, Discovery Sciences, R&D, AstraZeneca Gothenburg, Pepparedsleden 1,
431 50 Mölndal, Sweden.

## ABSTRACT

Accurate prediction of thermodynamic properties is essential in drug discovery
and materials science. Molecular dynamics (MD) simulations provide a principled
approach to this task, yet they typically rely on prohibitively long sequential sim-
ulations. Implicit Transfer Operator (ITO) Learning offers a promising approach
to address this limitation by enabling stable simulation with time steps orders of
magnitude larger than MD. However, to train ITOs, we need extensive, unbiased
MD data, limiting the scope of this framework. Here, we introduce Boltzmann
Priors for ITO (BoPITO) to enhance ITO learning in two ways. First, BoPITO
enables more efficient data generation, and second, it embeds inductive biases
for long-term dynamical behavior, simultaneously improving sample efficiency
by one order of magnitude and guaranteeing asymptotically unbiased equilibrium
statistics. Furthermore, we showcase the use of BoPITO in a new tunable sam-
pling protocol interpolating between ITOs trained on off-equilibrium simulations
and an equilibrium model by incorporating unbiased correlation functions. Code
is available at `https://github.com/olsson-group/bopito`.

## 1   INTRODUCTION

Efficient molecular dynamics (MD) simulation on long time-scales is critical to a large number of
scientific and engineering applications. Stable simulations rely on solvers taking tiny integration
time-steps, making the simulation of most phenomena impractical with current methods. Since
these simulations are stochastic, a simulation step corresponds to drawing a sample from a tran-
sition probability density $p(\mathbf{x}_{t+\tau} \mid \mathbf{x}_t)$, where $\tau$ is a tiny time-step. Recently, deep generative
models have emerged as a promising strategy to potentially speed up these simulations by learning
transition probability densities where $\tau$ is much larger (Schreiner et al., 2023; Klein et al., 2023;
Hsu et al., 2024; Fu et al., 2023) and thereby allow efficient sampling at long time-scales. Implicit
Transfer Operator (ITO) learning (Schreiner et al., 2023) learns such surrogate models at multiple
time-resolutions. While ITO has shown promise in accelerating simulations, it relies on extensive
unbiased simulation data which may not always be available. In Markov state models, this limita-
tion can be mitigated by integrating off-equilibrium simulations and enhanced sampling simulations
(Trendelkamp-Schroer et al., 2015; Rosta & Hummer, 2014; Wu et al., 2014; 2016). However, such
estimators are so far unavailable for deep generative surrogates of the transition density.

Here, we introduce Boltzmann Priors for Implicit Transfer Operator (BoPITO) learning (Figure 1).
BoPITO leverages pre-trained Boltzmann Generators (BG) (Noé et al., 2019; Viguera Diez et al.,
2024; Klein & Noé, 2024; Köhler et al., 2023; Köhler et al., 2020; Midgley et al., 2024; 2023) as
priors to enable data-efficient training of ITO models, leading to one order of magnitude reduction
in the simulation data needed for training. As the BG encodes the *invariant measure* or Boltzmann
distribution of the dynamics encoded by the transition density, BoPITO, by construction, guarantees
asymptotically unbiased equilibrium statistics. In this way, BoPITO can combine off-equilibrium
data and biased data encoded into a BG prior to train ITO models that predict MD across multiple
time-scales. Using BoPITO we introduce a new sampling strategy to recover approximate dynamics
from biased off-equilibrium data, a BG prior, and unbiased time-correlation data, providing a new
method for inverse problems for molecular systems.

---

[*]email: simonols@chalmers.se

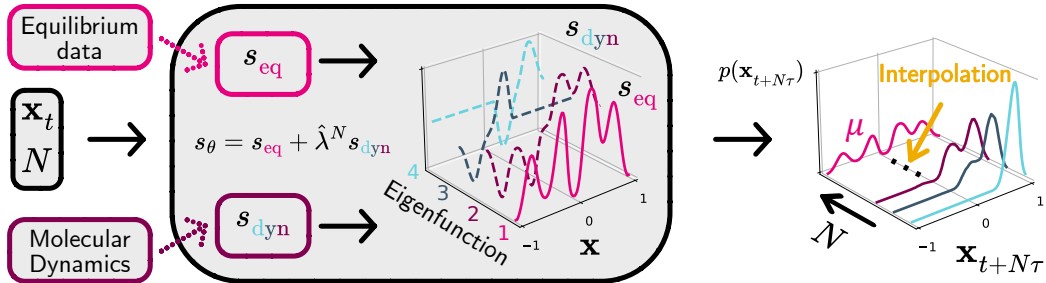

Figure 1: **Boltzmann Priors for Implicit Transfer Operators (BoPITO)** leverage pre-trained Boltzmann Generators to enable data-efficient training of surrogate models of the transition density after $N$ simulation steps. BoPITO is implemented using Score-Based Diffusion Models. The score of the model, $s_\theta$, separates the contributions from the first eigenfunction of the transfer operator (equilibrium density), $s_{\text{eq}}$, from the rest, $s_{\text{dyn}}$. BoPITO embeds inductive biases for long-term dynamical behavior and enables interpolation between off-equilibrium and equilibrium models.

Our main contributions are

1. **Boltzmann Priors for Implicit Transfer Operator Learning (BoPITO)**: We provide a principled way to leverage Boltzmann Generators as priors for Implicit Transfer Operator Learning, boosting sample efficiency, ensuring the generation of uncorrelated data and allowing asymptotically unbiased equilibrium statistics for $\tau \to \infty$.

2. **BoPITO interpolators**: A tunable sampling protocol facilitating interpolation of BoPITO models trained on off-equilibrium simulation data to an unbiased equilibrium distribution.

3. We show that BoPITO interpolators can **recover approximate dynamics from models trained on biased simulations**. We frame the optimization of interpolation parameters as an inverse problem and **select the interpolated ensemble that is most consistent with unbiased observables**.

## 2 BACKGROUND AND PRELIMINARIES

### 2.1 MOLECULAR DYNAMICS AND OBSERVABLES

Molecular dynamics (MD) is a widely used simulation method in chemistry, physics, and biology. It combines a mathematical model of the dynamics — e.g. Langevin dynamics (Langevin, 1908) — with a potential energy model, $U(\mathbf{x}) : \Omega \to \mathbb{R}$, of a system of interest, in turn providing detailed mechanistic insights of molecular systems, through the time evolution of particles in configuration space, $\mathbf{x} \in \Omega$. Practically, MD is solved by numerical integration, and time is discretized, with step $\tau$. Consequently, we can understand MD as a Markov process with a Normal transition density $p(\mathbf{x}_{t+\tau} \mid \mathbf{x}_t)$, whose associated Markov operator has the *Boltzmann distribution*,

$$\mu(\mathbf{x}) = \mathcal{Z}^{-1} \exp(-\beta U(\mathbf{x})), \text{ with } \mathcal{Z} = \int \mathrm{d}\mathbf{x} \, \exp(-\beta U(\mathbf{x})), \tag{1}$$

as its *invariant measure* or stationary distribution, where $\beta$ is the inverse temperature.

One important application of MD is to compute expectations or *observables* (Olsson, 2022):

1. **Stationary observables**:

$$O_a = \mathbb{E}_\mu \left[ a(\mathbf{x}) \right]. \tag{2}$$

2. **Dynamic observables / Time correlation functions**:

$$O_{a(t),b(t+N\tau)} = \mathbb{E}_{\mathbf{x}_t \sim \mu} \left[ \mathbb{E}_{\mathbf{x}_{t+N\tau} \sim p_\tau(\mathbf{x}_{t+N\tau} | \mathbf{x}_t)} \left[ a(\mathbf{x}_t) \cdot b(\mathbf{x}_{t+N\tau}) \right] \right], \tag{3}$$

where $p_\tau(\mathbf{x}_{t+N\tau}|\mathbf{x}_t)$ is the conditional probability density after $N$ simulation steps with time-step $\tau$. The maps $a$ and $b : \Omega \to \mathbb{R}^L$ serve as observable functions or *'forward models'* characterizing microscopic observation processes, e.g. indicating whether a drug is bound or not, or an interatomic distance, leading to observables such binding affinities and off-rates of a drug to a target protein.

Unfortunately, stable MD simulations rely on using time discretizations, $\tau$, on the order of $10^{-15}$ s, whereas properties such as protein folding or ligand unbinding occur on much longer time scales, $\sim 10^{-3} - 10^{-1}$ s. Due to the sequential nature of these simulations, an impractical number of simulation steps are necessary to predict these properties in an unbiased fashion. As a result, most MD data will be *'off-equilibrium'*, e.g. trajectories exploring one or a few of the modes of the Boltzmann distribution $\mu(\mathbf{x})$. See Appendix A.1 for more precise definitions of configuration space, off-equilibrium, unbiased and biased MD.

## 2.2 BOLTZMANN GENERATORS

Boltzmann Generators (BG) (Noé et al., 2019) are a generative machine learning approach to draw samples i.i.d. from the Boltzmann distribution, $\mu(\mathbf{x})$, of physical many-body systems (eq. 1). BGs learn a diffeomorphic map, $\mathcal{F}_{\boldsymbol{\theta}} : \mathbb{R}^N \to \mathbb{R}^N$, from a latent space, equipped with a simple base density $p(\mathbf{z})$, to configurational space of a physical system, such that the push-forward density $\hat{p}(\mathbf{x}) = \mathcal{F}_{\boldsymbol{\theta}} \# p(\mathbf{z})$ closely approximates the Boltzmann distribution (eq. 1). In practice, $\mathcal{F}_{\boldsymbol{\theta}}$ is implemented using an invertible neural network architecture with tractable Jacobian determinants to enable efficient sampling and exact sample likelihood computation (Chen et al., 2018; Papamakarios et al., 2021). BGs are trained either with approximately equilibrated simulation data or with biased simulations, from i.e. enhanced sampling, by employing appropriate reweighting (Ferrenberg & Swendsen, 1989; Shirts & Chodera, 2008). Unbiased samples can then be generated by importance re-sampling (Nicoli et al., 2020). Similarly, unbiased expectations can be computed using the importance weights $w(\mathbf{x}) = e^{-\beta U(\mathbf{x})}/\hat{p}(\mathbf{x})$. As such, BGs are *surrogates* of the target Boltzmann distribution, but do not model time-correlation statistics.

## 2.3 TRANSFER OPERATORS

Transfer operators (Ruelle, 1978; Schütte et al., 2009) provide a framework to describe the evolution of probability densities over time. Let $p$ denote an initial probability density function on $\Omega$, and $\rho$ its $\mu$-weighted version, $p = \mu\rho$. The Markov operator, $T_\Omega$, is defined using a transition density $p(\mathbf{x}_{t+\tau}|\mathbf{x}_t)$,

$$[T_\Omega(\tau) \circ \rho](\mathbf{x}_{t+\tau}) \equiv \frac{1}{\mu(\mathbf{x}_{t+\tau})} \int_\Omega \mu(\mathbf{x}_t)\rho(\mathbf{x}_t)p(\mathbf{x}_{t+\tau}|\mathbf{x}_t)\,\mathrm{d}\mathbf{x}_t. \tag{4}$$

This operator describes the $\mu$-weighted evolution of absolutely convergent probability density functions on $\Omega$ by a discrete-time increment $\tau$ given by the dynamics encoded in a transition density $p(\mathbf{x}_{t+\tau}|\mathbf{x}_t)$. In the context of molecular dynamics, $T_\Omega$ is a $\mu$-weighted equivalent to the Markov operator discussed above (sec. 2.1). The spectral form of the transfer operator is expressed as

$$[T_\Omega(\tau) \circ \rho](\mathbf{x}_{t+\tau}) = \sum_{i=1}^{\infty} \lambda_i(\tau)\langle\rho|\phi_i\rangle\,\psi_i(\mathbf{x}_{t+\tau}), \tag{5}$$

where $\lambda_i(\tau)$ are the eigenvalues, $\psi_i$ and $\phi_i$ are the corresponding right and left eigenfunctions, respectively, and $\langle f|g\rangle = \int_\Omega f(\mathbf{x})g(\mathbf{x})\,\mathrm{d}\mathbf{x}$. The eigenvalues $\lambda_i(\tau)$ depend on the parameter $\tau$ and are related to the characteristic relaxation rates $\kappa_i$ by $|\lambda_i(\tau)| = \exp(-\tau\kappa_i)$. The right and left eigenfunctions are related by the stationary density, such that $\phi_i(\mathbf{x}) = \mu(\mathbf{x})\psi_i(\mathbf{x})$. In reversible dynamics, all eigenvalues $\lambda_i$ are real and lie within the interval $-1 < \lambda_i \leq 1$. Notably, there is one eigenvalue $\lambda_1 = 1$, with corresponding eigenfunctions $\psi_1(\mathbf{x}) = \mathbf{1}$ and $\phi_1(\mathbf{x}) = \mu(\mathbf{x})$.

## 2.4 DIFFUSION MODELS

Diffusion models (DMs) (Ho et al., 2020; Song et al., 2020) are a popular generative modeling framework that approximates data densities $p(\mathbf{x}^0)$ by learning to invert a noising process (*'forward diffusion'*). The forward diffusion process is pre-specified and incrementally transforms the data distribution into a simple prior distribution, $p(\mathbf{x}^T)$, through simulation of a time-inhomogenous Markov process represented by the SDE,

$$\mathrm{d}\mathbf{x}^{t_{\mathrm{diff}}} = f(\mathbf{x}^{t_{\mathrm{diff}}}, t_{\mathrm{diff}})\,\mathrm{d}t_{\mathrm{diff}} + g(t_{\mathrm{diff}})\,\mathrm{d}W, \tag{6}$$

where $(0 < t_{\mathrm{diff}} < T)$ is the diffusion time, $f$ and $g$ are chosen functions, and $\mathrm{d}W$ is a Wiener process. To generate samples from the data distribution, $p(x^0)$, we can sample from $p(\mathbf{x}^T)$ and

solve the backward diffusion process (*'denoising process'*) (Anderson, 1982),

$$d\mathbf{x}^{t_{\text{diff}}} = \left[ f(\mathbf{x}^{t_{\text{diff}}}, t_{\text{diff}}) - g^2(t_{\text{diff}}) \nabla_{\mathbf{x}^{t_{\text{diff}}}} \log p(\mathbf{x}^{t_{\text{diff}}} | t_{\text{diff}}) \right] dt_{\text{diff}} + g(t_{\text{diff}}) dW. \quad (7)$$

This backward process is approximated using a deep neural network $\nabla_{\mathbf{x}^{t_{\text{diff}}}} \log p_{\boldsymbol{\theta}}(\mathbf{x}^{t_{\text{diff}}} | t_{\text{diff}}) = s_{\boldsymbol{\theta}}(\mathbf{x}^{t_{\text{diff}}}, t_{\text{diff}})$. The learned backward SDE can be recast into a probability flow ODE (Maoutsa et al., 2020; Song et al., 2021), which in turn can be interpreted as a continuous normalizing flow (Chen et al., 2018) which facilitates efficient sampling and enables sample likelihood evaluation.

## 2.5 IMPLICIT TRANSFER OPERATOR LEARNING

Implicit Transfer Operator (ITO) Learning (Schreiner et al., 2023) is a framework for learning surrogate models of the transition density, $p(\mathbf{x}_{N\tau} | \mathbf{x}_0)$, from MD data, where $N$ is an arbitrarily large integer. ITO leverages that the transfer operator framework allows us to express the transition density as

$$p(\mathbf{x}_{t+N\tau} | \mathbf{x}_t) = \sum_{i=1}^{\infty} \lambda_i^N(\tau) \psi_i(\mathbf{x}_t) \phi_i(\mathbf{x}_{t+N\tau}), \quad (8)$$

where $\psi_i$ and $\phi_i$ are independent of $\tau$ and $N$. See Appendix A.2 for a complete derivation.

This decomposition inspired a strategy to learn a conditional generative model $\mathbf{x}_{t+N\tau} \sim p_{\boldsymbol{\theta}}(\mathbf{x}_{t+N\tau} | \mathbf{x}_t, N)$ by sampling tuples $(\mathbf{x}_{t_i}, \mathbf{x}_{t_i+N_i\tau}, N_i)$ and training a generative model to mimic the empirical transition density at multiple time-horizons, $N\tau$. Here, we approximate ITO models with a conditional denoising diffusion probabilistic model (cDDPM) of the form

$$p(\mathbf{x}_{t+N\tau}^0 | \mathbf{x}_t, N) \equiv \int p(\mathbf{x}_{t+N\tau}^{0:T} | \mathbf{x}_t, N) \, d\mathbf{x}^{1:T}, \quad (9)$$

where $\mathbf{x}_{1:T}$ are latent variables of the same dimension as our output, and follow a joint density describing the backward diffusion process, eq. 7, and $\mathbf{x}^T \sim \mathcal{N}(0, \mathbb{I})$. We denote diffusion time in Diffusion Models using superscripts, while physical time is represented using subscripts. The conditional sample likelihood is given by

$$\ell(\mathbf{I}; \boldsymbol{\theta}) \equiv \prod_{i \in \mathbf{I}} p_{\boldsymbol{\theta}}(\mathbf{x}_{t_i+N_i\tau}^0 \mid \mathbf{x}_{t_i}, N_i) \quad (10)$$

where $\mathbf{I}$ is a list of generated indices $i$ specifying a time $t_i$ and a time-lag ($\tau$) integer multiple $N_i$, associating two time-points in a MD trajectory of length $M\tau$, $\mathbf{x} = \{\mathbf{x}_0, \mathbf{x}_\tau, \dots, \mathbf{x}_{(M-1)\tau}\}$.

Training is performed by optimizing an approximation of the variational bound of the log-likelihood (Ho et al., 2020),

$$\mathcal{L}(\boldsymbol{\theta}) = \mathbb{E}_{i \sim \mathbf{I}, \epsilon \sim \mathcal{N}(0, \mathbb{I}), t_{\text{diff}} \sim \mathcal{U}(0, T)} \left[ \| \epsilon - \hat{\epsilon}_{\boldsymbol{\theta}}(\widetilde{\mathbf{x}}_{t_i+N_i\tau}^{t_{\text{diff}}}, \mathbf{x}_{t_i}, N_i, t_{\text{diff}}) \|_2 \right], \quad (11)$$

where $\widetilde{\mathbf{x}}_t^{t_{\text{diff}}} = \sqrt{\bar{\alpha}^{t_{\text{diff}}}} \mathbf{x}_t + \sqrt{1 - \bar{\alpha}^{t_{\text{diff}}}} \epsilon$, with $\bar{\alpha}^{t_{\text{diff}}} = \prod_j^{t_{\text{diff}}} (1 - \beta_j)$ and $\beta_j$ is the variance of the forward diffusion process at diffusion time, $j$. $\hat{\epsilon}_{\boldsymbol{\theta}}(\cdot)$ is one of the two architectures presented by Schreiner et al. (2023). Following Arts et al. (2023) we express the score as

$$s_{\boldsymbol{\theta}}(\mathbf{x}_{t+N\tau}^{t_{\text{diff}}}, \mathbf{x}_t, N_i, t_{\text{diff}}) = -\frac{\hat{\epsilon}_{\boldsymbol{\theta}}(\widetilde{\mathbf{x}}_{t_i+N_i\tau}^{t_{\text{diff}}}, \mathbf{x}_{t_i}, N_i, t_{\text{diff}})}{\sqrt{1 - \bar{\alpha}^{t_{\text{diff}}}}}. \quad (12)$$

See Appendix A.3 for more details and pseudo-code on training and sampling algorithms.

## 3 RELATED WORK

**Sampling the Boltzmann distribution**   Apart from Boltzmann Generator-based approaches, there are a number of traditional ways to draw statistical samples from the Boltzmann distribution of molecular systems. Prominent examples include, molecular dynamics or Markov Chain Monte Carlo simulations coupled with enhanced sampling strategies (Hénin et al., 2022; Kamenik et al., 2022), including replica-based approaches (Earl & Deem, 2005; Sidler et al., 2016; Pasarkar et al., 2023), conformational flooding (Grubmüller, 1995), meta-dynamics (Laio & Parrinello, 2002), and

umbrella sampling (Torrie & Valleau, 1977), in particular when paired up with machine-learned collective variables (Chen & Ferguson, 2018; Wang et al., 2019b; Herringer et al., 2023; M. Sultan & Pande, 2017). The success of these approaches relies on choosing the right mechanism to enhance sampling across high free energy barriers, or low probability regions, for any given case (Carter et al., 1989). Finding such mechanisms typically involves substantial manual engineering of collective variables and other hyper-parameters. Finally, there are transition path sampling approaches (Dellago et al., 1998; Bolhuis et al., 2002), which are particularly powerful when combined with reinforcement learning (Jung et al., 2023) or deep generative priors (Plainer et al., 2023).

**Latent space simulators and Coarse graining**    Classical (fine-grained) molecular dynamics simulation can be coarse-grained beyond the Born-Oppenheimer approximation, such that atomic nuclei are merged together into 'beads.'(Noid, 2023) This strategy, in principle enables faster simulations due to the smaller number of particles and the acceleration of kinetics caused by the coarse-graining (CG) operation (Nüske et al., 2019). CG models can be estimated to closely approximate the thermodynamics of the corresponding fine-grained system through the optimization of two equivalent variational bounds (Noid et al., 2008; Lyubartsev & Laaksonen, 1995; Ercolessi & Adams, 1994). A relaxation of this bound was recently proposed to also allow for probabilistic reconstruction of fine-grained configurations (Chennakesavalu et al., 2023). These bounds have been used extensively to build CG force-field models (Husic et al., 2020; Wang et al., 2019a; Majewski et al., 2023; Charron et al., 2023) including implicit solvation models (Chen et al., 2021; Katzberger & Riniker, 2024), using deep neural networks due to their ability to capture multibody terms (Wang et al., 2021) to ultimately accelerate the prediction of equilibrium properties of molecular systems. Similarly, the development of *'latent space simulators'* where a learned, typically low-dimensional, latent space equipped either with a propagator (Sidky et al., 2020; Chennakesavalu et al., 2023; Wang et al., 2024), or not (Wang & Gómez-Bombarelli, 2019), is learned to enable efficient simulation. These approaches, in general aim to accelerate molecular simulations akin to BoPITO, yet due to the CG operation, the molecular dynamics (kinetics) will be accelerated, and detailed knowledge of the unbiased dynamics are needed to correct this (Nüske et al., 2019; Crommelin & Vanden-Eijnden, 2011). Concurrent work, presents MDGen where conformational states are tokenized and in turn used to generate multiple frames of a MD trajectory jointly (Jing et al., 2024).

**Transfer Operator surrogates**    Analysis of MD data often involves building transfer operator surrogates such as Markov state models (MSM) (Schütte et al., 1998; Prinz et al., 2011; Swope et al., 2004; Husic & Pande, 2018), time-lagged independent components analysis (Molgedey & Schuster, 1994; Ziehe & Müller, 1998; Pérez-Hernández et al., 2013), Markov field models or dynamic graphical models, (Olsson & Noé, 2019; Mardt et al., 2022; Hempel et al., 2022), VAMPnets (Mardt et al., 2018; Wu & Noé, 2019), or observable operator models (Wu et al., 2015). Markov state models are time-space discrete approximations of the transfer operator and Deep Generative MSM (Wu et al., 2019) and VAMPnets (Mardt et al., 2018) learn the space discretization through deep neural networks. Dynamic graphical models or Markov field models (Hempel et al., 2022) represent a time-space discrete approximation of the transfer operator injecting a (conditional) independence assumption of molecular subsystems, when modeling the transition probability, leading to better scaling for systems with poor time-scale separation (Olsson & Noé, 2019). Apart from ITO (Schreiner et al., 2023), several other deep generative approaches for modeling the transition density of molecular dynamics have recently proposed. *Timewarp* where a normalizing flow is used to encode the transition density (Klein et al., 2023) with limited transferability to enable metropolized sampling of unbiased equilibrium distributions (Hastings, 1970). *Score dynamics* use a DDPM to model the displacements of an initial configuration towards a time-lagged one, achieving picosecond time-steps simulation and limited transferability (Hsu et al., 2024). However, unlike in the context of MSMs (Trendelkamp-Schroer et al., 2015; Rosta & Hummer, 2014; Wu et al., 2014; 2016), there are no deep generative transition density surrogates available leveraging available information about the equilibrium distribution — BoPITO is one such method.

## 4    BOLTZMANN PRIORS FOR IMPLICIT TRANSFER OPERATOR LEARNING

ITO training requires extensive, unbiased MD simulations to capture the statistical distribution of rare events. This data-intensive requirement can hinder ITO's practical implementation. Furthermore, models trained on off-equilibrium simulations, which may exhibit non-representative statistics, can lead to inaccurate predictions, compromising their utility in downstream applications. In practice, however, both unbiased off-equilibrium simulations and information about the equilibrium distribution from potentially biased simulations (Hénin et al., 2022) can be cheaply generated, and both encode information about the dynamical behavior of molecular systems. We introduce Boltzmann Priors for Implicit Transfer Operator (BoPITO), as a learning paradigm for ITO models that leverage available information about the equilibrium distribution.

BoPITO uses pre-trained models of the equilibrium distribution to improve ITO learning in four ways. First, it helps ensure broad sampling of $\Omega$ proportional to the equilibrium distribution for subsequent MD simulations yielding information about the transition density $p(\mathbf{x}_\tau \mid \mathbf{x}_0)$ from across $\Omega$. Second, we use it to fix the stationary part of the learned transition density, boosting the sample efficiency when learning models of molecular dynamics. Third, it imposes an inductive bias of long-time dynamical behavior allowing for recovery of unbiased Boltzmann distribution for long-time horizons. Fourth, using BoPITO, we introduce a novel tunable sampling protocol interpolating ITO models trained on off-equilibrium simulation data and an unbiased equilibrium distribution.

### 4.1    EFFICIENT DATA GENERATION

Training data for ITO models consists of several independent unbiased MD simulations. Following the adaptive sampling strategy (Bowman et al., 2010; Doerr & De Fabritiis, 2014; Viguera Diez et al., 2024; Betz & Dror, 2019), used extensively in the molecular dynamics simulation community, we use a pre-trained BG to generate initial conditions to simulations ensuring broad sampling across $\Omega$ proportional to the Boltzmann distribution. As long as these trajectories reach a 'local equilibrium' we can in principle recover an unbiased model of the molecular dynamics (Nüske et al., 2017), albeit without relying on running one or a few very long simulations to reach the global equilibrium.

### 4.2    LONG-TERM DYNAMICS INDUCTIVE BIAS FOR ITO

One important application of ITO is to allow for one-step sampling of long-time-scale dynamics. However, real datasets often contain a limited number of effective samples for long time-scales, leading to potential biases in models. To mitigate this issue, we propose separating the equilibrium contribution from the time-dependent components:

$$p(\mathbf{x}_{t+N\tau}|\mathbf{x}_t) = \mu(\mathbf{x}_{t+N\tau}) + \sum_{i=2}^{\infty} \lambda_i^N(\tau)\phi_i(\mathbf{x}_{t+N\tau})\psi_i(\mathbf{x}_t), \qquad (13)$$

where we have used that $\lambda_1 = 1$ and its corresponding eigenfunctions are $\phi_1 = \mu$ and $\psi_1 = \mathbf{1}$. Additionally, we introduce a decay in the time-dependent component, creating an inductive bias that asymptotically samples from an available equilibrium model for long-term dynamics. Using the spectral decomposition of the transition density (eq. 13), we choose the score model as

$$s(\mathbf{x}_{t+N\tau}^{t_{\text{diff}}}, \mathbf{x}_t, N, t_{\text{diff}}, \boldsymbol{\theta}) = s_{\text{eq}}(\mathbf{x}_{t+N\tau}^{t_{\text{diff}}}, t_{\text{diff}}) + \hat{\lambda}^N s_{\text{dyn}}(\mathbf{x}_{t+N\tau}^{t_{\text{diff}}}, \mathbf{x}_t, N, t_{\text{diff}}, \boldsymbol{\theta}), \qquad (14)$$

where $s_{\text{eq}}(\mathbf{x}_{\text{diff}}^t, t_{\text{diff}})$ is the score of a pre-trained surrogate of the equilibrium distribution model, $0 < \hat{\lambda} < 1$ is a hyper-parameter and $s_{\text{dyn}}(\mathbf{x}_{t+N\tau}^{t_{\text{diff}}}, \mathbf{x}_t, N, t_{\text{diff}}, \boldsymbol{\theta})$ accounts for the time-dependent components. As $N \to \infty$, $s_{\text{eq}}$ dominates and the model samples from the equilibrium model, see Appendix A.4 for an example. By interpreting the score field as a velocity field, we can reformulate the DM as a continuous normalizing flow which we can use for Metropolized sampling of the unbiased equilibrium distribution (Klein et al., 2023). The proposed factorization is principled, corresponding to a separation of the score of the transition density into a stationary and dynamic part, where the first part is considered known, see Appendix A.5 for details. In practice, we first train $s_{\text{eq}}$ (if not provided) using equilibrium data. Then we train $s_{\text{dyn}}$ with unbiased, possibly off-equilibrium, MD data while keeping $s_{\text{eq}}$ fixed and we choose $\hat{\lambda}$ performing hyper-parameter optimization, see Appendix A.6 for details. We discuss alternative formulations of the score in Appendix A.7.

### 4.3 BoPITO interpolators

A significant issue in MD simulations is the unbiased sampling of transitions over high free-energy barriers, e.g. channels in $\mu(\mathbf{x})$ with very little probability mass, typically coinciding with mixing on $\Omega$. This difficulty leads to an inability to predict time-correlation statistics for large $N\tau$. With BoPITO we can define an interpolation between models trained on off-equilibrium simulations and the equilibrium distribution. In this manner, we can approximate dynamics not seen explicitly in the data. For a time-dependent model, $s_{\text{dyn}}$, trained on off-equilibrium simulations with maximum training lag, $N_{\text{max}}$, we define a BoPITO interpolator as a model with the following score function:

$$s(\mathbf{x}^{t_{\text{diff}}}, \mathbf{x}_t, N_{\text{int}}, t_{\text{diff}}) = s_{\text{eq}}(\mathbf{x}^{t_{\text{diff}}}, t_{\text{diff}}) + \hat{\lambda}^{N_{\text{int}}} s_{\text{dyn}}(\mathbf{x}^{t_{\text{diff}}}, \mathbf{x}_t, N_{\text{max}}, t_{\text{diff}}), \tag{15}$$

with $N_{\text{int}} \geq N_{\text{max}}$. This defines an interpolator because $N_{\text{int}} = N_{\text{max}}$ generates samples from the model distribution with maximum lag and $N_{\text{int}} \to \infty$ generates samples from the equilibrium model.

Inspired by methods in molecular biophysics (Kolloff & Olsson, 2023; Olsson et al., 2017; Bottaro & Lindorff-Larsen, 2018; Salvi et al., 2016), we propose to choose the interpolation parameter, $N_{\text{int}}$, as the most consistent with an unbiased dynamical observable, such as experimental data. That is, for a lag $N > N_{\text{max}}$ and an unbiased dynamic observable $O_N^*$, we choose

$$N_{\text{int},N} = \underset{N_{\text{int}}}{\arg\min} |O_N^* - O_{N_{\text{int}}}|, \tag{16}$$

where $O_{N_{\text{int}}}$ is the dynamic observable estimated with samples from the BoPITO interpolator. This way BoPITO can integrate off-equilibrium and equilibrium MD with experimental data. In practice, we find the interpolator to generate some high-energy states. However, we can alleviate the high-energy structures by alternating long-lag interpolation steps with short-lag non-interpolation steps for local relaxation. We discuss how BoPITO interpolators alleviate out-of-distribution issues in Appendix A.8.

## 5 Results

For detailed parameters of the experiments below, we refer to Appendix A.9.

### 5.1 Systems

**Prinz potential** is a 1D potential commonly used for benchmarking MD sampling methods (Prinz et al., 2011). We set the observable functions, $a$ and $b$ in eq. 3, to be the identity function for computing dynamic observables. For details, see Appendix A.10.

**Alanine Dipeptide** is a small peptide with 22 atoms. We use publicly available data from Dibak et al. (2022), containing $1\,\mu$s simulation time split in 20 trajectories. Simulation is performed in an implicit solvent with $2\,\text{fs}$ integration time-step, and data is saved every $1\,\text{ps}$. We choose,

$$a(\mathbf{x}) = b(\mathbf{x}) = \begin{bmatrix} \sin\phi(\mathbf{x}) \\ \cos\phi(\mathbf{x}) \end{bmatrix}, \tag{17}$$

where $\phi(\mathbf{x})$ is a torsion angle involved in the slowest transition in the system. For details, see Appendix A.11.

**Chignolin** (cln025) is a fast folding protein with 10 residues, 166 atoms and 93 heavy atoms. We use molecular dynamics data previously reported by Lindorff-Larsen et al. (2011). The data is proprietary but available upon request for research purposes. The simulations were performed in explicit solvent with a $2.5\,\text{fs}$ time-step and the positions was saved at $200\,\text{ps}$ intervals. We extract all heavy atoms positions from the simulations and train models on this data. We use the fraction of native contacts Lindorff-Larsen et al. (2011) to define a dynamic observable with,

$$a(\mathbf{x}) = b(\mathbf{x}) = \frac{\sum_{i=1}^{N_{\text{res}}} \sum_{j>i}^{N_i} \frac{1}{1+e^{10(d_{ij}(\mathbf{x})-d_{ij}^*-1)}}}{\sum_{i=1}^{N_{\text{res}}} N_i}, \tag{18}$$

where the first sum in the numerator iterates over all $N_{\text{res}}$ residues in the protein, while the second sum considers the $N_i$ native contacts of residue $i$ separated by at least seven residues in the primary sequence. Here, $d_{ij}(\mathbf{x})$ and $d_{ij}^*$ represent the $C_\alpha - C_\alpha$ distances of residues $i$ and $j$ in the structure $\mathbf{x}$ and the native structure, respectively. This observable quantifies the protein's foldedness, with values ranging from 0 (unfolded) to 1 (folded). For a detailed explanation, refer to Appendix A.12.

## 5.2 BOLTZMANN PRIORS FOR TRAINING DATA GENERATION

Modeling the transition density over $\Omega$ requires observing transitions across the state space. In the usual setting, only one or a few initial conditions are available when data collection starts. Here, we explore the case where a BG is available before data collection and compare it against the baseline, where only one initial condition is known. We use a BG trained on equilibrium data of the Prinz potential to sample the initial conditions of our training trajectories. We also generate trajectories using a single starting point (crystal, $\mathbf{x} = 0.75$) and compare their performance with long simulations (Figure 2). We compare the two different data generation strategies using the metric, $|\Delta\text{correlation}|$, which measures the absolute difference of the time-correlation function (dynamic observable) compared to the MD ground truth (Appendix A.13). We find that the performance of models using Boltzmann priors for data generation is superior for different lag times (Figure 2a) and number of trajectories (Figure 2b). The gap between BG and the crystal baseline increases with lag-time, and as expected decreases with the number of generated trajectories.

## 5.3 BOPITO EFFICIENTLY SAMPLES LONG-TERM DYNAMICS IN A LOW-DATA CONTEXT

Fixing the equilibrium contribution to the score field effectively reduces the number of parameters that need to be estimated. To test whether this prior information manifests as an improved sample efficiency we compared ITO and BoPITO models against each other with varying sizes of training data.

We find that the BoPITO models achieve a higher accuracy for long-term dynamics compared to ITO models when data is scarce, as the equilibrium distribution is known *a priori* and does not need to be learned from simulation data (Figure 3). The inductive bias in BoPITO models enables them to learn long-term dynamics, even in scenarios where ITO models fail. For the Prinz Potential, we find that while ITO suffers from poor performance modeling long-term dynamics when data is scarce,

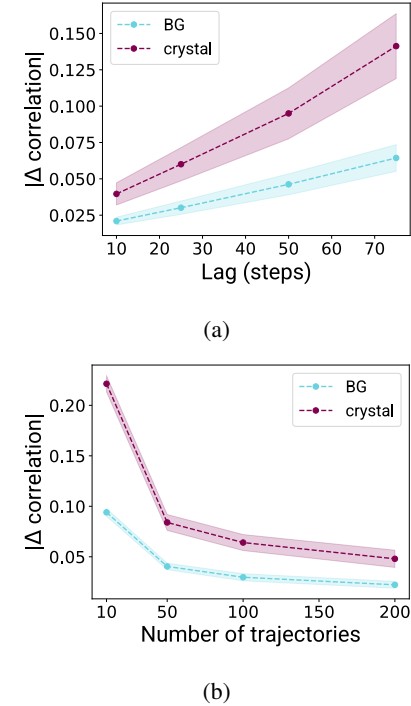

(a)

(b)

Figure 2: Absolute difference in correlation with respect to long unbiased MD simulations (lower is better) of models trained on trajectories initialized on samples from a Boltzmann Generator (BG) and a single structure (crystal, $\mathbf{x} = 0.75$) for the Prinz Potential under direct sampling. The former presents superior performance for different lag times (number of trajectories = 50) (a) and number of trajectories (b). The shaded areas correspond to 95 % confidence interval.

BoPITO models accurately capture long-term dynamics without worsening the performance on short and medium time-scales. The results for Alanine Dipeptide and Chignolin show favorable scaling with increasing system size: we find that an order of magnitude more data is needed to train an ITO to the same accuracy as a BoPITO model trained on the same data. Moreover, we show in Appendix A.14 that the energies of configurations sampled from our prior Boltzmann Generator used in our experiments match closely the energies of MD samples.

## 5.4 INTERPOLATING BETWEEN MODELS TRAINED ON OFF-EQUILIBRIUM DATA AND THE BOLTZMANN DISTRIBUTION WITH EXPERIMENTAL DATA

In practical settings, our MD data will be off-equilibrium, e.g. having sampled only one or a few of the relevant modes in the Boltzmann distribution $\mu(\mathbf{x})$ in a given trajectory. Consequently, unless extensive data across the domain $\Omega$ can be collected, models based on such data will be biased. An alternative to collecting more simulation data is to use a multi-modal strategy where experimental data is used to fill the gaps left by simulation data and bridge to long time-scale dynamics (Salvi et al., 2016; Kolloff & Olsson, 2023). In this section we explore the potential of BoPITO interpolators to integrate dynamic observables with off-equilibrium simulation data to recover unbiased long-term dynamics.

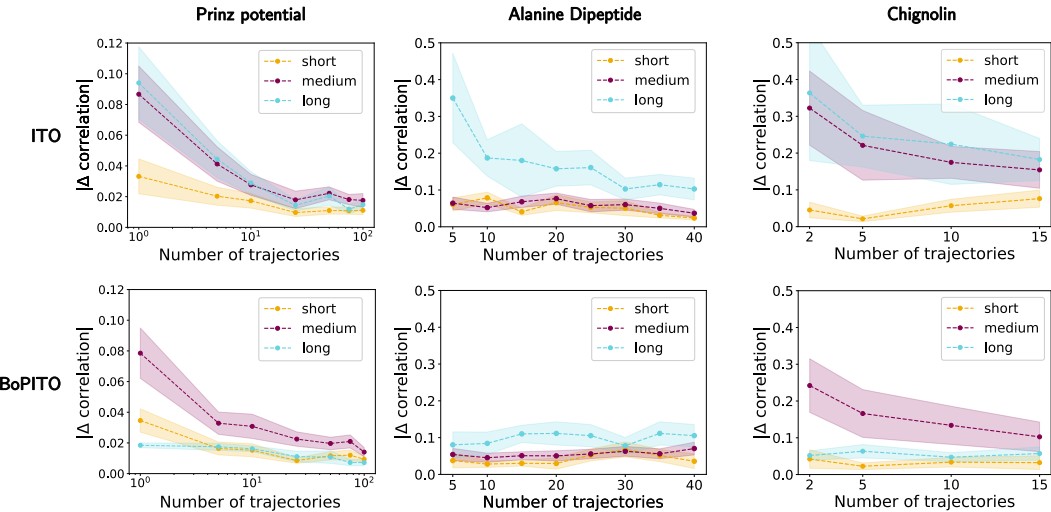

Figure 3: Absolute difference in correlation with respect to long unbiased MD simulations (lower is better, one-step sampling) of ITO (left) and BoPITO (right) split into short, medium, and long time-scales against the number of training trajectories for the Prinz potential (top) and Alanine Dipeptide (middle) and Chignolin (bottom). The shaded areas correspond to a 95 % confidence interval.

We showcase BoPITO interpolators on Alanine Dipeptide. By removing the transitions between the modes of the Boltzmann distribution corresponding to the slowest process in the system, e.g. $\phi$ crossing 0 or 2, we generate a biased simulation data, resembling a realistic scenario. We then train ITO and BoPITO models using these biased trajectories. For lags $> 100$, we sample the BoPITO interpolator estimated by matching to an unbiased dynamic observable defined by eq. 26, allowing us to overcome a systematic error in the correlation function observed in the biased MD data and for an ITO model trained on these data (Figure 4).

Beyond reproducing the provided dynamic observable, the interpolator also demonstrates a remarkable ability to capture the underlying microscopic dynamics (Figure 5). Even with excellent agreement, the generated ensembles only slightly overestimates density in the transition state region. However, this effect could easily be alleviated by annealing a short MD simulation to the interpolation as was recently shown (Viguera Diez et al., 2024).

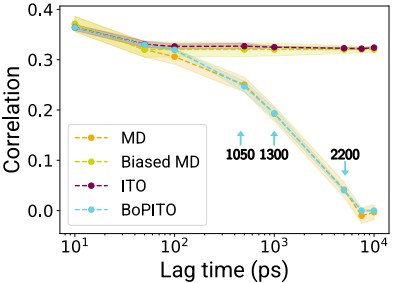

Figure 4: Fitting an interpolator of a biased BoPITO model for Alanine Dipeptide. Both the ITO and BoPITO models are trained on biased MD simulations of Alanine Dipeptide. We fit a BoPITO interpolator selecting the most consistent ensemble with the unbiased correlation function specified in eq. 16. Numbers with blue arrows specify the interpolation parameter, $N_{\text{int}}$.

## 6 LIMITATIONS AND FUTURE WORK

**Choice of hyper-parameter $\hat{\lambda}$** The hyper-parameter $\hat{\lambda}$, defines a global relaxation or mixing time-scale of the dynamics after which the model is guaranteed to sample equilibrium. In Appendix A.6 we describe a protocol to determine a bound for this parameter. However, developing a similar protocol for a transferable BoPITO model would likely require modifications to accommodate the diversity of global relaxation time-scales across different systems.

**No Chapman-Kolmogorov Guarantee** When training on biased or off-equilibrium data where we rely on establishing ergodicity through interpolation we cannot guarantee self-consistency of the dynamics in the Chapman-Kolmogorov sense.

**Surrogate model** BoPITO inherits the current limitations of ITO, such as generalization over chemical space and thermodynamic variables, and scaling. Furthermore, current models cannot guarantee unbiased sampling dynamics for non-equilibrium ensembles, which would require closed-form expressions for the target path probabilities.

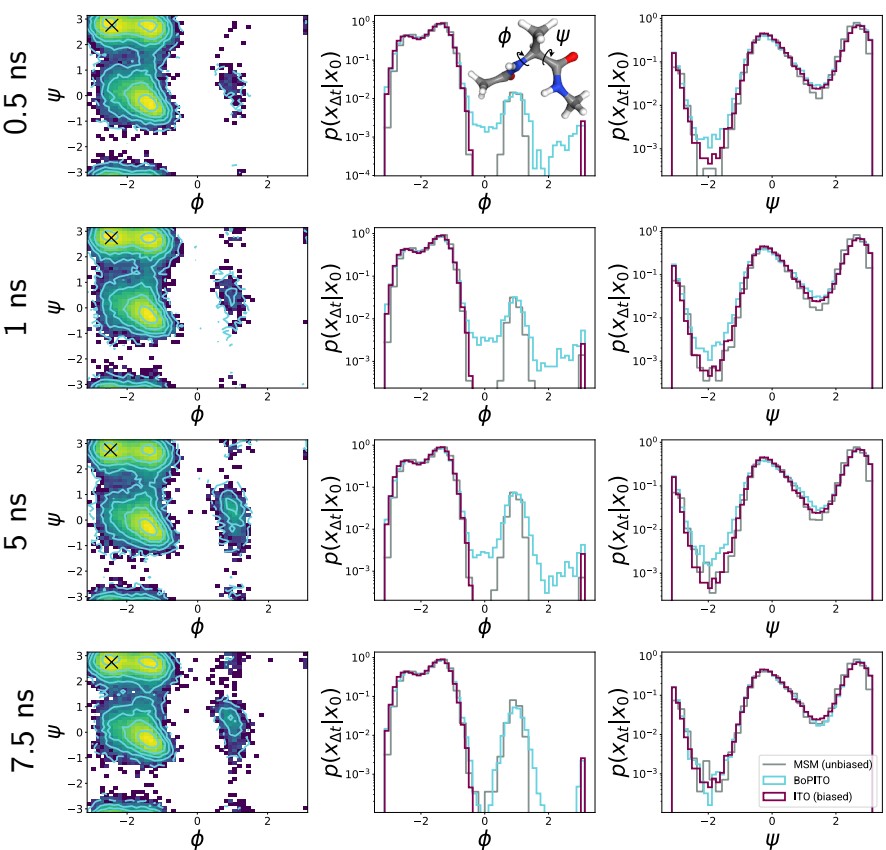

Figure 5: BoPITO can incorporate unbiased dynamic observables to correct a model trained on biased data. Rows of increasing time-lag (from top to bottom). Contour plots correspond to a BoPITO interpolator. The first column shows conditional transition densities projected onto the torsion angles $\phi$ and $\psi$ (inset). The black cross indicates the initial condition. The second and third columns show marginal distributions of $\phi$ and $\psi$, respectively. MSM stands for a Markov State Model of the unbiased MD data.

## 7 CONCLUSION

We introduce Boltzmann Priors for Implicit transfer Operator Learning (BoPITO), a framework to enhance ITO learning in three ways. First, a broad sampling of configuration space is used to initialize short off-equilibrium MD simulations. Second, we parameterize the transition density as an interpolation towards a pre-trained Boltzmann Generator, improving sample efficiency by an order of magnitude. BoPITO is a principled approach to embedding prior knowledge of the stationary distribution of Markovian dynamics as an inductive bias for long-term dynamical behavior. Third, our approach enables interpolation between models trained on off-equilibrium data and the equilibrium distribution, and we can recover accurate models of unseen dynamics when informed by unbiased observables. Consequently, BoPITO is the first method to allow for the integration of multiple sources of information into the generation of deep generative surrogates of molecular dynamics.

ACKNOWLEDGMENTS

The authors thank Prof. Rocío Mercado and Prof. Fredrik Johansson for their useful feedback on the manuscript. This work was partially supported by the Wallenberg AI, Autonomous Systems and Software Program (WASP) funded by the Knut and Alice Wallenberg Foundation. The computations in this work were enabled by the Berzelius resource provided by the Knut and Alice Wallenberg Foundation at the National Supercomputer Centre.

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

# A APPENDIX

## A.1 DEFINITIONS

- **Configuration space**: Mathematical space in which all possible states or positions of a physical system are represented. For example, in a classical MD simulation, the positions of all the atoms in the simulation.

- **Unbiased simulations**: Simulations performed with standard MD, i.e. Langevin Dynamics. They are unbiased because they generate samples from the underlying transition density. Particularly, for asymptotically large simulation times, unbiased simulations sample the Boltzmann distribution $\mu(\mathbf{x})$. However, because of energy barriers in the potential energy landscape, unbiased simulations may not explore some modes of the Boltzmann distribution if the simulation time is not long enough. Therefore, generating samples with unbiased MD often leads to off-equilibrium data.

- **Off-equilibrium data**: Simulation trajectories whose underlying statistics do not represent well those of the equilibrium distribution. One example is lack of ergodicity: when the simulation fails to explore some modes in the Boltzmann distribution. Even if having explored the full state space, it is possible to have off-equilibrium trajectories because of rare events, which require to be sampled "many" times to accurately represent equilibrium statistics.

- **Biased simulation**: Simulations performed with modified versions of naive MD that speed up the exploration of state space. These methods are a subset of enhanced sampling and include meta-dynamics (Laio & Parrinello, 2002), replica exchange (Earl & Deem, 2005) and others (Sidler et al., 2016; Pasarkar et al., 2023). Data generated with enhanced sampling do not resemble the underlying transition density, but, often, can be re-weighted to represent the equilibrium distribution.

## A.2 EIGEN DECOMPOSITION OF THE TRANSFER OPERATOR AND TRANSITION DENSITY

The two-fold composition of the transfer operator, eq. 5, acting on an initial $\mu$-weighted density, $\rho$, is

$$[T_\Omega^2(\tau) \circ \rho](\mathbf{x}_{t+2\tau}) = \sum_{i=1}^\infty \lambda_i(\tau) \left\langle \sum_{j=1}^\infty \lambda_j(\tau)\langle\rho|\phi_j\rangle \, \psi_j|\phi_i \right\rangle \, \psi_i(\mathbf{x}_{t+2\tau})$$

$$= \sum_{i=1}^\infty \lambda_i(\tau) \left( \sum_{j=1}^\infty \lambda_j(\tau)\langle\rho|\phi_j\rangle \, \langle\psi_j|\phi_i\rangle \right) \, \psi_i(\mathbf{x}_{t+2\tau})$$

$$= \sum_{i=1}^\infty \lambda_i^2(\tau)\langle\rho|\phi_i\rangle \, \psi_i(\mathbf{x}_{t+2\tau}),$$

where we have used the orthonormality of the eigenfunctions, that is $\langle\psi_j|\phi_i\rangle = \delta_{ij}$. Similarly,

$$[T_\Omega^N(\tau) \circ \rho](\mathbf{x}_{t+N\tau}) = \sum_{i=1}^\infty \lambda_i^N(\tau)\langle\rho|\phi_i\rangle \, \psi_i(\mathbf{x}_{t+N\tau}). \tag{19}$$

The transition density can be re-written in terms of the spectral decomposition of the transfer operator by choosing the initial density, $p$, as a Dirac delta function $\delta_{\mathbf{x}_t}(\mathbf{x})$, that is $\rho(\mathbf{x}) = \frac{\delta_{\mathbf{x}_t}(\mathbf{x})}{\mu(\mathbf{x})}$.

Then,

$$
\begin{aligned}
p(\mathbf{x}_{t+N\tau}|\mathbf{x}_t) &= \mu(\mathbf{x}_{t+N\tau}) \left[ T_\Omega^N(\tau) \circ \frac{\delta_{\mathbf{x}_t}}{\mu} \right] (\mathbf{x}_{t+N\tau}) \\
&= \mu(\mathbf{x}_{t+N\tau}) \sum_{i=1}^\infty \lambda_i^N(\tau) \left\langle \frac{\delta_{\mathbf{x}_t}}{\mu} \middle| \phi_i \right\rangle \psi_i(\mathbf{x}_{t+N\tau}) \\
&= \sum_{i=1}^\infty \lambda_i^N(\tau) \left\langle \delta_{\mathbf{x}_t}(\mathbf{x})|\psi_i \right\rangle \psi_i(\mathbf{x}_{t+N\tau}) \, \mu(\mathbf{x}_{t+N\tau}) \\
&= \sum_{i=1}^\infty \lambda_i^N(\tau) \psi_i(\mathbf{x}_t) \phi_i(\mathbf{x}_{t+N\tau}).
\end{aligned}
$$

### A.3 IMPLICIT TRANSFER OPERATOR DETAILS

Implicit Transfer Operator (ITO) Learning (Schreiner et al., 2023) is a framework for learning surrogate models of the transition density, $p(\mathbf{x}_{N\tau}|\mathbf{x}_0)$. ITO models are trained sampling tuples $(\mathbf{x}_{t_i}, \mathbf{x}_{t_i+N_i\tau}, N_i)$ and training a generative model to mimic the empirical transition density at multiple time-horizons, $N\tau$. Training is done following Algorithm 1.

---

**Algorithm 1** Training. DisExp is defined in Algorithm 4

---

**Input:** $n$ MD-trajectories; $\mathcal{X} = \{\mathbf{x}_0^j, \dots, \mathbf{x}_{t_j}^j\}_{j=0}^n$, ITO score-model; $\hat{\epsilon}_\theta$, max lag; $N_{\max}$
$\mathcal{X}' = \text{Concatenate}(\{\mathbf{x}_0^j, \dots, \mathbf{x}_{t_j-N_{\max}}^j\}_{j=0}^n)$
**while** not converged **do**
    $\mathbf{x}_t \sim \text{Choice}(\mathcal{X}')$
    $N \sim \text{DisExp}(N_{\max})$
    $t_{\text{diff}} \sim \text{Uniform}(0, T)$
    Take gradient step on:
    $\nabla_{\boldsymbol{\theta}} \left[ \|\epsilon - \hat{\epsilon}_{\boldsymbol{\theta}}(\widetilde{\mathbf{x}}_{t+N\tau}^{t_{\text{diff}}}, \mathbf{x}_t, N, t_{\text{diff}})\|_2 \right]$
**end while**
**return** $\hat{\epsilon}_{\boldsymbol{\theta}}$

---

Once a model is trained, it can be sampled by following Algorithm 2.

---

**Algorithm 2** Sampling from $\hat{p}_{\boldsymbol{\theta}}(\mathbf{x}_0, N)$

---

**Input:** initial condition $\mathbf{x}_0$, lag; $N$, diffusion steps; $T_{\text{diff}}$, ITO score-model; $\hat{\epsilon}_\theta$
$\mathbf{x}_N^{T_{\text{diff}}} \sim \mathcal{N}(\mathbf{0}, \mathbf{1})$
**for** $t_{\text{diff}} = T_{\text{diff}} \dots 1$ **do**
    $\epsilon \sim \mathcal{N}(\mathbf{0}, \mathbf{1})$
    $\mathbf{x}_N^{t_{\text{diff}}-1} = \frac{1}{\sqrt{\alpha^{t_{\text{diff}}}}} \left( \mathbf{x}_N^{t_{\text{diff}}} - \frac{1-\alpha^{t_{\text{diff}}}}{\sqrt{1-\bar{\alpha}^{t_{\text{diff}}}}} \hat{\epsilon}_\theta(\mathbf{x}_N^{t_{\text{diff}}}, \mathbf{x}_0, N, t_{\text{diff}}) \right) + \sqrt{\beta_t}\epsilon$
**end for**
**return** $\mathbf{x}_N^0$

---

Several sampling steps can be annealed to sample longer lag-times as depicted in Algorithm 3.

---

**Algorithm 3** Ancestral sampling. Sampling from $p_{\boldsymbol{\theta}}$ is defined in Algorithm 2

---

**Input:** initial condition $\mathbf{x}_0$, lag $N$, ancestral steps $n$.
**Allocate** $\mathcal{T} \in \mathbb{R}^{(n+1) \times \dim(\mathbf{x}_0)}$
$\mathcal{T}[0] = \mathbf{x}_0$
**for** $i = 1 \dots n$ **do**
$\quad \mathbf{x}_i \sim \hat{p}_{\boldsymbol{\theta}}(\mathcal{T}[i-1], N)$
$\quad \mathcal{T}[i] = \mathbf{x}_i$
**end for**
**return** $\mathcal{T}$

---

---

**Algorithm 4** Sampling from DisExp

---

$N_{\log} \sim \mathrm{Uniform}(0, \log(N_{\max}))$
**Return:** $\mathrm{floor}(\exp(N_{\log}))$

---

BoPITO uses Algorithm 1 for training and Algorithm 2 for sampling as well, but uses the score mode in eq. 14.

### A.4 DECAY OF TIME-DEPENDENT SCORE TERM

In Figure 6, we show the average time-dependent component of the score,

$$s_N = \mathbb{E}_{i \sim \mathbf{I}, t_{\mathrm{diff}} \sim \mathcal{U}(0,T)} \left[ |\hat{\lambda}^N s_{\mathrm{dyn}}(\mathbf{x}_{t_i + N\tau}^{t_{\mathrm{diff}}}, \mathbf{x}_{t_i}, N, t_{\mathrm{diff}}, \boldsymbol{\theta})| \right], \tag{20}$$

of a trained BoPITO model of the Prinz Potential with $\hat{\lambda} = 0.994$. The expectation for $\mathbf{I}$ corresponding to the sampling of tuples as during training (Alg. 1). The model remains flexible for small lags but eventually decreases to 0, sampling from the equilibrium model.

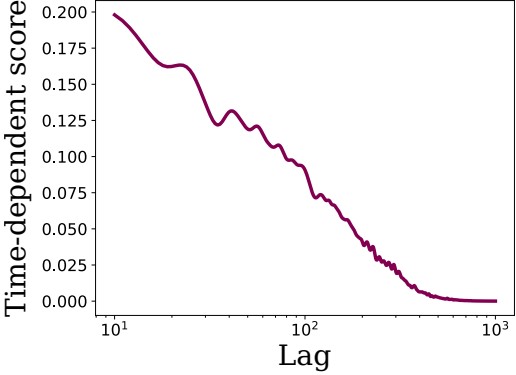

Figure 6: Average time-dependent component of the score, eq. 20, of a trained model BoPITO model of the Prinz Potential.

## A.5 THE SCORE OF THE TRANSITION PROBABILITY AND BOPITO

The score of the transition probability can be written as

$$
\begin{aligned}
\nabla_{\mathbf{x}_{N\tau}} \log p(\mathbf{x}_{N\tau}|\mathbf{x}_0, N) &= \sum_i^\infty \lambda_i(\tau)^N \frac{\psi_i(\mathbf{x}_0)}{p(\mathbf{x}_{N\tau}|\mathbf{x}_0, N)} \nabla_{\mathbf{x}_{N\tau}} \phi_i(\mathbf{x}_{N\tau}) \\
&= \frac{\nabla_{\mathbf{x}_{N\tau}} \mu(\mathbf{x}_{N\tau})}{p(\mathbf{x}_{N\tau}|\mathbf{x}_0, N)} + \sum_{i=2}^\infty \lambda_i(\tau)^N \frac{\psi_i(\mathbf{x}_0)}{p(\mathbf{x}_{N\tau}|\mathbf{x}_0, N)} \nabla_{\mathbf{x}_{N\tau}} \phi_i(\mathbf{x}_{N\tau}) \\
&= \frac{\nabla_{\mathbf{x}_{N\tau}} \mu(\mathbf{x}_{N\tau})}{\mu(\mathbf{x}_{N\tau})} \frac{\mu(\mathbf{x}_{N\tau})}{p(\mathbf{x}_{N\tau}|\mathbf{x}_0, N)} \\
&\quad + \hat{\lambda}^N \sum_{i=2}^\infty \left( \frac{\lambda_i(\tau)}{\hat{\lambda}} \right)^N \frac{\psi_i(\mathbf{x}_0)}{p(\mathbf{x}_{N\tau}|\mathbf{x}_0, N)} \nabla_{\mathbf{x}_{N\tau}} \phi_i(\mathbf{x}_{N\tau}).
\end{aligned}
$$

$\frac{\nabla_{\mathbf{x}_{N\tau}} \mu(\mathbf{x}_{N\tau})}{\mu(\mathbf{x}_{N\tau})} = \nabla_{\mathbf{x}_{N\tau}} \log \mu(\mathbf{x}_{N\tau})$ is the score of the Boltzmann distribution and can be modeled with the score of a Boltzmann Generator, $s_{\mathrm{eq}}(\mathbf{x}^{t_{\mathrm{diff}}}, t_{\mathrm{diff}})$. $\frac{\mu(\mathbf{x}_{N\tau})}{p(\mathbf{x}_{N\tau}|\mathbf{x}_0, N)} \to 1$ as $N \to \infty$ and can be modeled with a scalar neural network $1 + \hat{\lambda}^N f_{\boldsymbol{\theta}}(\mathbf{x}_{N\tau}^{t_{\mathrm{diff}}}, \mathbf{x}_0, N, t_{\mathrm{diff}})$. The last term can be modeled with $g_{\boldsymbol{\theta}}(\mathbf{x}_{N\tau}^{t_{\mathrm{diff}}}, \mathbf{x}_0, N, t_{\mathrm{diff}})$. The corresponding score model is

$$
\begin{aligned}
s_{\boldsymbol{\theta}}(\mathbf{x}_{N\tau}^{t_{\mathrm{diff}}}, \mathbf{x}_0, N, t_{\mathrm{diff}}) &= s_{\mathrm{eq}}(\mathbf{x}^{t_{\mathrm{diff}}}, t_{\mathrm{diff}})(1 + \hat{\lambda}^N f_{\boldsymbol{\theta}}(\mathbf{x}_{N\tau}^{t_{\mathrm{diff}}}, \mathbf{x}_0, N, t_{\mathrm{diff}}) + \hat{\lambda}^N g_{\boldsymbol{\theta}}(\mathbf{x}_{N\tau}^{t_{\mathrm{diff}}}, \mathbf{x}_0, N, t_{\mathrm{diff}}) \\
&= s_{\mathrm{eq}}(\mathbf{x}^{t_{\mathrm{diff}}}, t_{\mathrm{diff}}) + \hat{\lambda}^N \underbrace{\left( s_{\mathrm{eq}}(\mathbf{x}^{t_{\mathrm{diff}}}, t_{\mathrm{diff}}) f_{\boldsymbol{\theta}}(\mathbf{x}_{N\tau}^{t_{\mathrm{diff}}}, \mathbf{x}_0, N, t_{\mathrm{diff}}) + g_{\boldsymbol{\theta}}(\mathbf{x}_{N\tau}^{t_{\mathrm{diff}}}, \mathbf{x}_0, N, t_{\mathrm{diff}}) \right)}_{s_{\mathrm{dyn}}(\mathbf{x}_{N\tau}^{t_{\mathrm{diff}}}, \mathbf{x}_0, N, t_{\mathrm{diff}})}.
\end{aligned}
$$

We can aggregate $f$ and $g$ to a single neural network component, $s_{\mathrm{dyn}}(\mathbf{x}_{N\tau}^{t_{\mathrm{diff}}}, \mathbf{x}_0, N, t_{\mathrm{diff}})$, to get

$$
s_{\boldsymbol{\theta}}(\mathbf{x}_{N\tau}^{t_{\mathrm{diff}}}, \mathbf{x}_0, N, t_{\mathrm{diff}}) = s_{\mathrm{eq}}(\mathbf{x}^{t_{\mathrm{diff}}}, t_{\mathrm{diff}}) + \hat{\lambda}^N s_{\mathrm{dyn}}(\mathbf{x}_{N\tau}^{t_{\mathrm{diff}}}, \mathbf{x}_0, N, t_{\mathrm{diff}}). \tag{21}
$$

For simplicity, we incorporate the structure of the transition density into the diffusion model, not only for $t_{\mathrm{diff}} = 0$ (the learned data distribution), but for all $t_{\mathrm{diff}} > 0$. We do not observe this choice to limit model expressivity in our experiments.

## A.6 FITTING $\hat{\lambda}$

The hyperparameter $\hat{\lambda}$ controls the time-scale at which the BoPITO model transitions to sampling the equilibrium model. Ideally, $\hat{\lambda}$ should be similar to the eigenvalue corresponding with the slowest process in the system, $\lambda_2$. However, its value is not generally available and requires extensive unbiased simulation data to be accurately estimated. If $\hat{\lambda}$ is too small, the model may prematurely relax to equilibrium, limiting its ability to capture non-equilibrium dynamics. Conversely, if $\hat{\lambda}$ is too large, the benefits of the BoPITO framework could diminish. Therefore, careful tuning of $\hat{\lambda}$ can be crucial for optimal performance.

As illustrated in Figure 7, grid-search hyper-parameter tuning can select an appropriate $\hat{\lambda}$ for a single-system model. Too small values of $\hat{\lambda}$ can hinder the model's ability to capture long-time-scale dynamics, leading to increased loss. We recommend choosing the smallest $\hat{\lambda}$ that yields a plateau in the $\hat{\lambda}$-loss curve (elbow rule), as demonstrated in Figure 7 (b) for the Prinz potential. However, practitioners should be cautious about sampling longer lags than the implied time-scale defined by $\hat{\lambda}$ if they cannot guarantee the system relaxes to equilibrium for those time-scales. Fitting curves for Alanine Dipeptide and Chignolin are shown in Figure 8.

## A.7 ALTERNATIVE SCORE FORMULATIONS

The score in eq. 14 is principled and resembles the score of the transition density. However, in practice, we found that this score can lead to suboptimal performance in learning short time-scales if the dynamic component fails to dominate the equilibrium component for small $N$. We only observed this for Chignolin. We relate this issue to numerical limitations due to very different scale

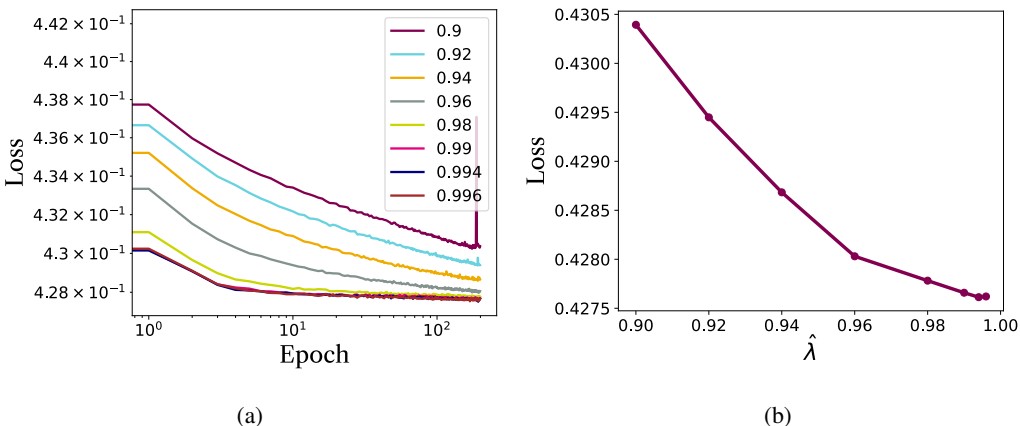

(a)                                                    (b)

Figure 7: Average loss against training epochs for the Prinz potential for different values of $\hat{\lambda}$ (a) and average final loss versus $\hat{\lambda}$ (b). Averages are taken w.r.t. 10 runs. Final losses are computed as the average between epochs 190 and 200. The greatest eigenvalue of the system under our simulation parameters is $0.994$.

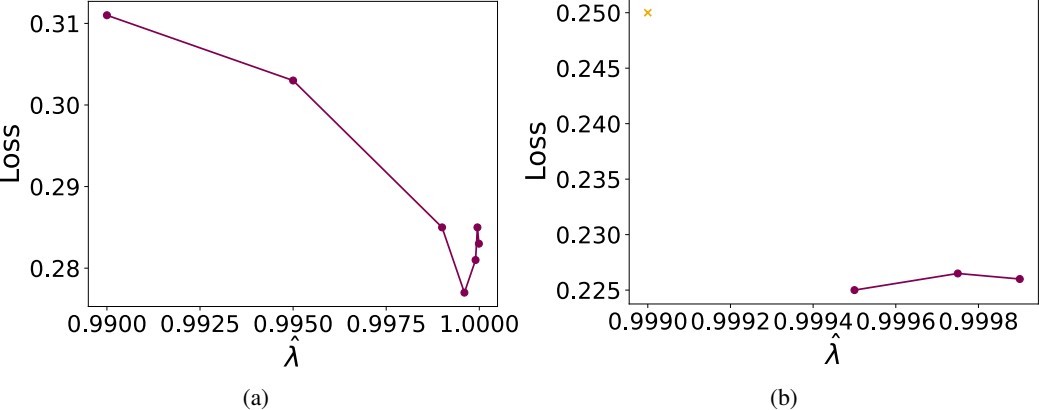

(a)                                                    (b)

Figure 8: Average final loss against different values of $\hat{\lambda}$ for Alanine Dipeptide (a) and Chignolin (b). The smallest value of $\hat{\lambda}$ that allowed numerically stable training for Chignolin was $0.9995$. Orange cross symbolizes that the trainings crashed because of numerical instability reasons.

in neural network weights for different values of $N$. We remark that short time-scales are the least relevant since they can be easily sampled with MD. Still, to mitigate this effect we propose the following alternative factorization,

$$s_{\boldsymbol{\theta}}(\mathbf{x}_{N\tau}^{t_{\text{diff}}}, \mathbf{x}_0, N, t_{\text{diff}}) = f(N)s_{\text{eq}}(\mathbf{x}^{t_{\text{diff}}}, t_{\text{diff}}) + \hat{\lambda}^N s_{\text{dyn}}(\mathbf{x}_{N\tau}^{t_{\text{diff}}}, \mathbf{x}_0, N, t_{\text{diff}}), \qquad (22)$$

where $f(N)$ is an increasing function and tends to 1 for large $N$. One potential option that does not introduce extra hyper-parameters is

$$s_{\boldsymbol{\theta}}(\mathbf{x}_{N\tau}^{t_{\text{diff}}}, \mathbf{x}_0, N, t_{\text{diff}}) = \frac{\hat{\lambda}^{-2N} - \hat{\lambda}^{2N}}{\hat{\lambda}^{-2N} + \hat{\lambda}^{2N}}s_{\text{eq}}(\mathbf{x}^{t_{\text{diff}}}, t_{\text{diff}}) + \hat{\lambda}^N s_{\text{dyn}}(\mathbf{x}_{N\tau}^{t_{\text{diff}}}, \mathbf{x}_0, N, t_{\text{diff}}). \qquad (23)$$

Moreover, to ensure adequate convergence to the equilibrium distribution and avoid over-fitting to unbiased MD for long time-scales, we use the following decay score for $N > N_{\text{decay}}$,

$$s_{\boldsymbol{\theta}}(\mathbf{x}_{N\tau}^{t_{\text{diff}}}, \mathbf{x}_0, N, t_{\text{diff}}) = s_{\text{eq}}(\mathbf{x}^{t_{\text{diff}}}, t_{\text{diff}}) + \hat{\lambda}^N s_{\text{dyn}}(\mathbf{x}_{N\tau}^{t_{\text{diff}}}, \mathbf{x}_0, N_{\text{decay}}, t_{\text{diff}}). \qquad (24)$$

We use this version for the experiments conducted on Chignolin and set $N_{\text{decay}} = 3t_2$, where $t_2$ is the unitless characteristic time of the slowest process in the system, $t_2 = \frac{-1}{\log \lambda_2}$.

## A.8 How BoPITO interpolators mitigate out-of-distribution effects

BoPITO interpolators mitigate out-of-distribution effects by, first, fixing $N = N_{max}$ as the argument of $s_{dyn}$. This choice ensures that none of our neural networks are evaluated outside of the training domain. Second, we alter long-lag interpolation steps with short-lag non-interpolation steps for local relaxation. Since we in practice see that that the long-lag steps occasionally generates 'off data manifold' states, e.g. structures with potential high energies, the local 'relaxation' steps projects us back onto the data manifold. In practice, just one short-lag non-interpolation step is sufficient to overcome this. Third, we use dynamic observables to fit the interpolation parameter. Without any additional source of information we don't know how to choose $N_{int}$. However, when we use dynamic observables, we can select the interpolated ensemble which is the most consistent with experimental observables. So this helps calibrate the time-scales to stay "in distribution".

## A.9 Experimental parameters

### A.9.1 Boltzmann Priors for dataset generation

For different numbers of trajectories, $n$, we train $5000/n$ ITO models on $n$ Prinz potential trajectories of length 150. Trajectories do not overlap among different trainings. The maximum model lag is 100 steps.

### A.9.2 BoPITO samples efficiently long-term dynamics in data-sparse scenarios

**Prinz potential** 10 models are trained on non-overlapping trajectory sets for different numbers of trajectories. The trajectories length is $10,000$ and the maximum model lag is $1,000$. We consider lags of $10, 25$, and $50$ as short, $100, 200$, and $300$ as medium, and $500, 750$, and $1,000$ as long time-scales in our experiments, see Figure 10 (b) for a visual reference. $\hat{\lambda}$ is set to $0.994$ for all BoPITO experiments.

**Alanine Dipeptide** 10 models are trained on potentially overlapping random trajectory sets for different numbers of trajectories. The trajectories length is $12,500$ and the maximum model lag is $10,000$. We consider lags of $5, 10$, and $50$ as short, $100, 500$, and $1000$, as medium, and $2500, 7500$, and $10000$ as long time-scales in our experiments, see Figure 11 (b). $\hat{\lambda}$ is set $0.9996$ for all BoPITO models.

**Chignolin** 4 models are trained on potentially overlapping random trajectory sets for different numbers of trajectories. The trajectories length is $35,000$ and the maximum model lag is $30,000$. We consider lags of $10, 50$, and $100$ as short, $500, 1000$, and $5000$, as medium, and $10000, 20000$, and $30000$ as long time-scales in our experiments, see Figure 13 (b). $\hat{\lambda}$ is set $0.9995$ for all BoPITO models. BoPITO models are trained using the score in eq. 23 and the decay function in eq. 24.

| Diffusion steps | 500 | | Diffusion steps | $1,000$ |
|---|---|---|---|---|
| Noise schedule | Sigmoidal | | Noise schedule | Polynomial |
| Batch size | $2,097,152$ | | Batch size | $1,024/32$ |
| Learning rate | 0.001 | | Learning rate | 0.001 |
| Layers | 3 | | Score layers | 5 |
| Embedding dimension | 256 | | Embedding layers | 2 |
| Net dimension | 256 | | n_features | 64 |
| Optimizer | Adam | | Optimizer | Adam |
| Inference ODE steps | 50 | | Inference ODE steps | $100/50$ |

Table 1: Architectural and training parameters of MB-ITO models (a) and SE3-ITO (b). Batch sizes and inference ODE steps refer to Alanine Dipeptide and Chignolin experiments respectively.

### A.9.3 INTERPOLATING BETWEEN MODELS TRAINED ON OFF EQUILIBRIUM DATA AND THE BOLTZMANN DISTRIBUTION

We remove the transitions resembling the slowest process in the system, $\phi$ crossing 0 or 2, to generate biased simulation data. When a transition occurs, we remove one frame before and after the transition and split the trajectory. We train both biased ITO and BoPITO models with a maximum model lag of 100 on the resulting biased dataset. For lags $> 100$, we sample the BoPITO interpolator fitted on the unbiased dynamic observable defined by eq. 26, see Figure 4. We perform interpolation by sampling with the score model in eq. 15 followed by one round of non-interpolation sampling with lag 100 steps for local relaxation.

### A.9.4 ARCHITECTURAL, TRAINING AND INFERENCE DETAILS

We use MB-ITO as the architecture of models for the Prinz potential, and SE3-ITO, an $SE(3)-$equivariant neural network, for Alanine Dipeptide, both introduced in Schreiner et al. (2023). We report architectural and training hyper-parameters in Table 1. Models are trained until convergence in the log-log loss plot.

### A.10 PRINZ POTENTIAL

The Prinz potential is a 1D potential commonly used for benchmarking MD sampling methods. The potential is defined as

$$U(x) = 4\left(x^8 + 0.8e^{-80x^2} + 0.2e^{-80(x-0.5)^2} + 0.5e^{-40(x+0.5)^2}\right). \tag{25}$$

We generate trajectories using an Euler-Maruyama integrator using the library Deeptime (Hoffmann et al., 2021). We set the integrator time-step to $1 \cdot 10^{-5}$, and the temperature, mass, and damping factor to 1. In Figure 9 we show histograms of the position of a particle after $N$ steps, starting from 0.75, and in Figure 10 (a) we report an implied time-scales plot. We use the identity function to define a dynamic observable/correlation function of this system and show it in Figure 10 (b).

### A.11 ALANINE DIPEPTIDE

Alanine Dipeptide is a small peptide with 22 atoms. We use publicly available data from Dibak et al. (2022), containing a total of $1 \mu s$ simulation time split in 20 trajectories. Simulation is performed in an implicit solvent with $2 fs$ integration time-step and data is saved every $1 ps$. We choose,

$$a(x) = b(x) = \begin{bmatrix} \sin\phi \\ \cos\phi \end{bmatrix}, \tag{26}$$

to define a dynamic observable of this system. The torsion angle $\phi$ is involved in the slowest transition observed in the simulation, see Figure 12. We combine these vectors taking the inner product for computing dynamic observables. We show this correlation function in Figure 11 (a), and implied time-scales plot in Figure 11 (b) and a histogram of the torsions $\phi$ and $\psi$ aggregating all simulation data in Figure 12.

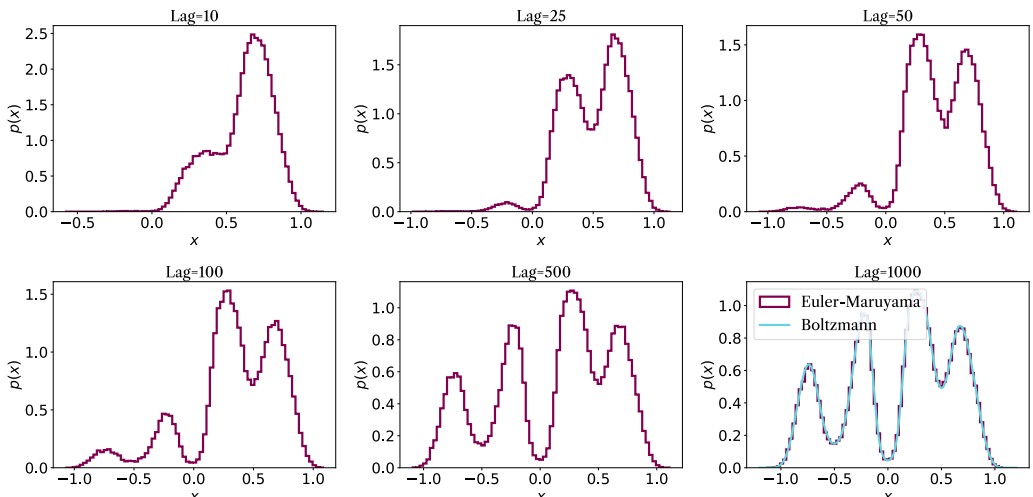

Figure 9: Conditional density of a simulation of the Prinz potential starting at $x = 0.75$ for different lags (steps). Long-term dynamics approaches the Boltzmann distribution.

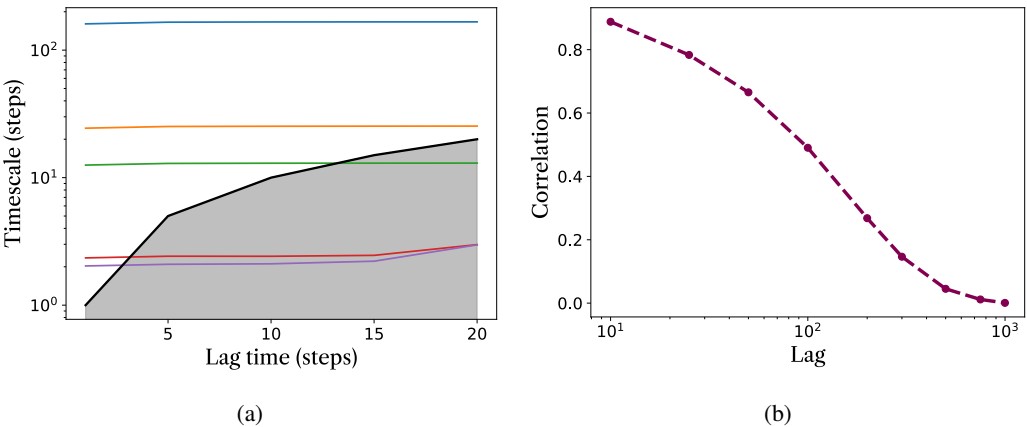

Figure 10: Implied time-scales (a) and dynamic observable under the identity function (b) for the Prinz Potential. Errors in (b) are smaller than the dots. In (a), the implied time-scales of the 5 slowest processes in the system are computed for different lags. The color order, from longest to shortest implied time-scale, is blue, orange, green, red, and purple.

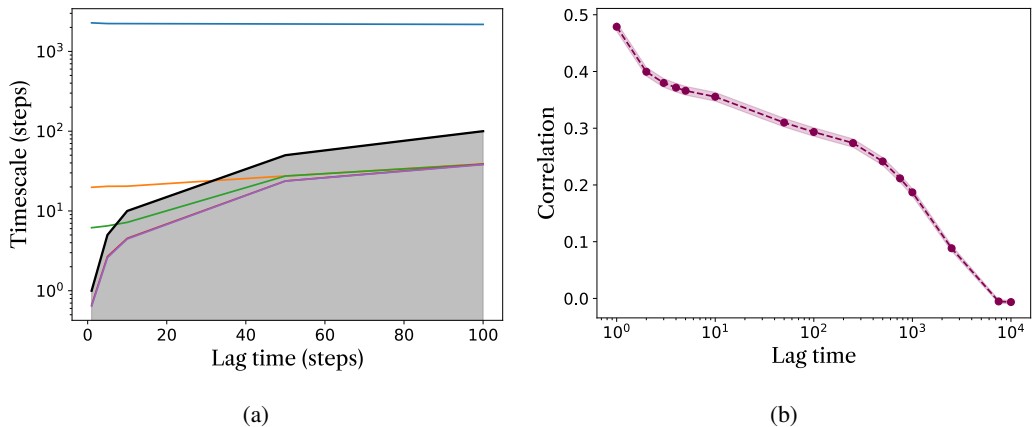

(a)                                                              (b)

Figure 11: Implied time-scales for Alanine Dipeptide computed with a Markov State Model on torsional angles $\phi$ and $\psi$ (a) and dynamic observable in eq. 26 (b). In (a), the implied time-scales of the 5 slowest processes in the system are computed for different lags. The color order, from longest to shortest implied time-scale, is blue, orange, green, red, and purple.

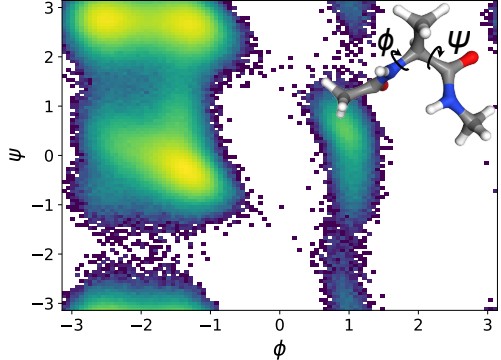

Figure 12: Histogram of the torsional angles $\phi$ and $\psi$ of Alanine Dipeptide (insert). Data is aggregated among all trajectories in the dataset introduced in Dibak et al. (2022).

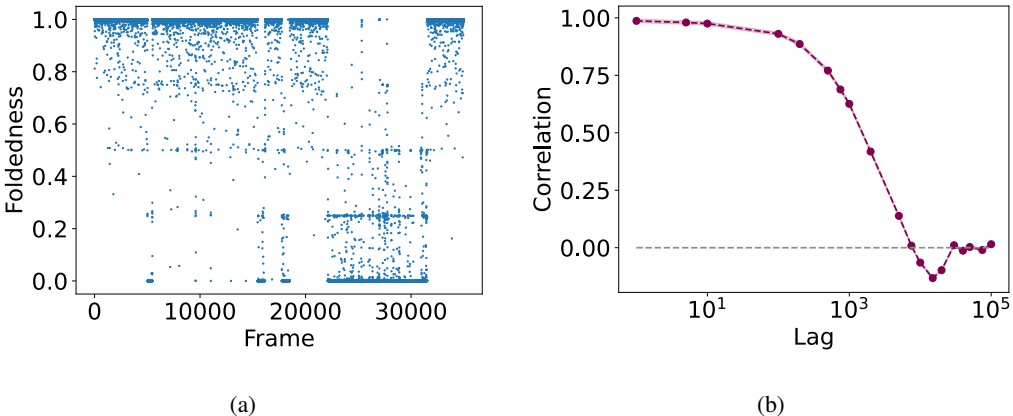

(a)              (b)

Figure 13: Time evoluation of 'foldedness' in a subset of the reference MD simulation (a) and dynamic observable in eq. 27 (b).

### A.12 CHIGNOLIN

Chignolin (cln025) is a fast folding protein with 10 residues, 166 atoms and 93 heavy atoms. We use molecular dynamics data previously reported by Lindorff-Larsen et al. (2011). The data is proprietary but available upon request for research purposes. The simulations were performed in explicit solvent with a 2.5 fs time-step and the positions was saved at 200 ps intervals. We extract all heavy atoms positions from the simulations and train models on this data. We use the reaction coordinate proposed in Lindorff-Larsen et al. (2011) as to define a dynamic observable,

$$a(\mathbf{x}) = b(\mathbf{x}) = \frac{\sum_{i=1}^{N_{\text{res}}} \sum_{j>i}^{N_i} \frac{1}{1+e^{10(d_{ij}(\mathbf{x}) - d_{ij}^* - 1)}}}{\sum_{i=1}^{N_{\text{res}}} N_i}, \tag{27}$$

where the first sum in the numerator iterates over all $N_{\text{res}}$ residues in the protein, while the second sum considers the $N_i$ native contacts of residue $i$ separated by at least seven residues in the primary sequence. We define native contacts as residue pairs separated by at least seven residues in the primary sequence and with $C_\alpha$ atoms closer than 10 Å in the native structure. Moreover, $d_{ij}(\mathbf{x})$ and $d_{ij}^*$ represent the distances between the $C_\alpha$ atoms of residues $i$ and $j$ in the structure $\mathbf{x}$ and the native structure, respectively. This observable quantifies the protein's foldedness, with values ranging from 0 (unfolded) to 1 (folded). In Figure 13 (a) we show the evolution of eq. 27 on a subset of the reference simulation data and we observe how the protein undergoes transformations between the folded and unfolded states. In Figure 13 (b) we visualize the corresponding dynamic observable.

### A.13 METRICS

We evaluate models computing differences in the dynamic observables introduced in section 5.1 w.r.t. long MD simulations,

$$|\Delta\text{correlation}|_N = |O_N^* - O_N^{\text{model}}|, \tag{28}$$

where $O_N^*$ is the normalized observable predicted by MD for lag $N$ and $O_N^{\text{model}}$ is the model's prediction. See Appendix A.15 for details on normalization. We report the average difference over different training runs.

### A.14 ENERGIES FOR ALANINE DIPEPTIDE

In Figure 14 we observe a remarkable agreement in energy densities of samples generated with our Boltzmann Generator and MD. GBSAOBForce is the implicit solvent model energy.

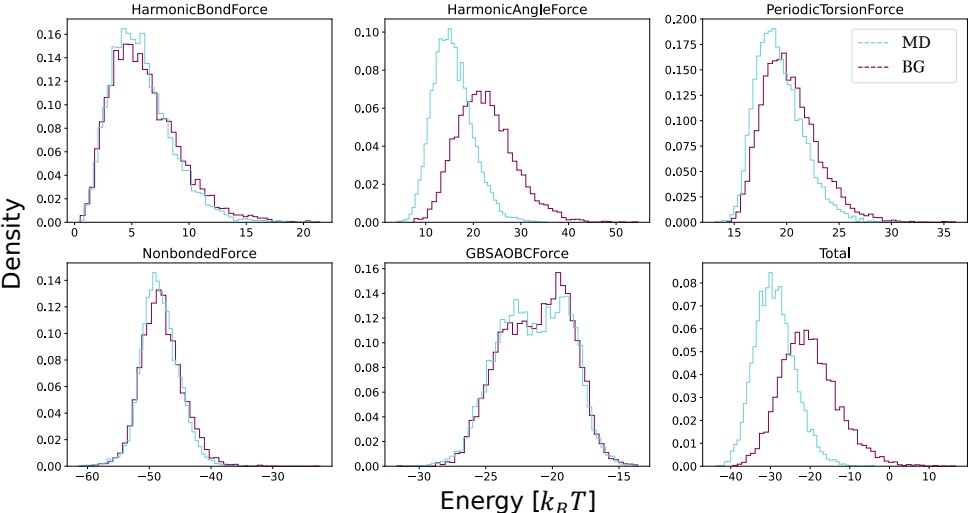

Figure 14: Energy and energy components of samples generated with a Boltzmann Generator (BG) and MD.

### A.15 CORRELATION FUNCTION NORMALIZATION

We subtract the mean and divide by correlation at lag $0$ to normalize our dynamic observables. Subtracting the mean guarantees that the correlation function asymptotically decays to $0$, and we divide by the correlation at lag $0$ so that correlation can be read as a fraction of correlation at time $0$. The resulting normalized correlation function is

$$\frac{E[(f(\mathbf{x}_t) - E[f(\mathbf{x}_t)]) \left(g(\mathbf{x}_{t+\Delta t}) - E[g(\mathbf{x}_{t+\Delta t})]\right)]}{E[(f(\mathbf{x}_t) - E[f(\mathbf{x}_t)]) \left(g(\mathbf{x}_t) - E[g(\mathbf{x}_t)]\right)]}. \tag{29}$$

When comparing different methods, we compute $E[f(\mathbf{x}_t)]$, $E[g(\mathbf{x}_{t+\Delta t})]$ and $E[(f(\mathbf{x}_t) - E[f(\mathbf{x}_t)]) \left(g(\mathbf{x}_t) - E[g(\mathbf{x}_t)]\right)]$ using long MD simulations and use these same normalizing factors for all methods.

