# OpenReview forum: "Boltzmann priors for Implicit Transfer Operators"
_ICLR.cc/2025/Conference — ICLR 2025 Poster_

### Official Review · Reviewer_hu2r · 2024-10-27

**Soundness:** 3
**Presentation:** 3
**Contribution:** 4
**Rating:** 8
**Confidence:** 4

**Summary:**

The paper uses a pretrained Boltzmann Generator to include un-biased equilibrium information into Implicit Transfer Operators (ITOs).
ITOs learn "time coarse grained" transition densities from molecular dynamics (MD) data.
The challenge is that ITOs require (a) unbiased (b) large quanities of MD data to train.
Boltzmann Generators (BG) can approximate the true Boltzmann distribution even if trained with biased data when combined with importance resampling.


The paper suggest three ways to improve ITOs with Boltzmann generators:
1. Sampling of conformations proportional to the equilibrium distribution from the Boltzmann Generator as starting points for trajectories
2. Inductive bias: Separate learning of the transition density into a fixed stationary part (the Boltzmann generator) and learned time-dependent component to improve sample efficiency.
(Based on the spectral decomposition of the transfer operator, where the equilibrium distribution is an eigenfunction with an eigenvalue of one.) An explicit temporal decay of the time-dependent component ensures convergence to the unbiased Boltzmann distribution for long-time horizons.
3. Approximate rare dynamics outside the data, by interpolating between ITOs trained on off-equilibrium data (time-dependent component) and the unbiased equilibrium distribution (Boltzmann generator).

**Strengths:**

* Temporal coarse-graining of molecular dynamics is of very high relevance to simulate long-horizon observables more efficiently
* Making use of cheaply obtainable off-equilibrium data is a very promising approach to train more scalable models
* Method seems sound and well-founded in physical/statistical principles
* The results look promising, although somewhat limited due to the small scale experiments

**Weaknesses:**

* The paper would benefit from clearer writing and more extensive background sections. It seems to assume that readers are already familiar with the implicit transfer operators framework, which will often not be the case. More explanations and pseudocode (even if it is just in the appendix) would make the paper much easier to read.
* The experiments are conducted only on small-scale systems, and it is unclear how well this approach would scale to more realistic scenarios.

**Questions:**

* The formalism decomposes the operator in a sum of left and right eigenfunctions. Are these eigenfunctions used anywhere in the training objectives directly, or are they solely used to justify/derive the form of the score function ansatz?
* In line 139 and 142 it says $\lambda_i = exp(-\tau \kappa_i)$ but also that $-1 < \lambda_i < 1$. How is it possible that $\lambda_i = exp(-\tau \kappa_i) < 0$ though?
* I am having some trouble understanding the interpolators section. N_max is the maximum training lag, but then we sample with N_int > N_max. Doesn't that cause out-of-distribution problems? Maybe you could add a bit more explanation in the appendix as to why the given model implies an interpolator.
* line 60: How exactly do you define "off-equilibrium" and "biased"?
* line 191 and following: what is $t_{diff}$?
* Looking at Figure 3, it seems like ITO and BoPITO have very similar performance on short and medium time scales, and BoPITO mostly outperforms ITO on long timescales, presumably due to the enforcement of the Boltzman distribution for long time horizons. I am wondering how often it is precisely the medium time scales that we are interested in though?

## Suggestions
* Include a section that explains the ITO framework in more detail and includes pseudo-code
* line 107 and other places, I find formulations like "few of the modes of μ(x)" are easier to read if written as  "few of the modes of the *Boltzmann distribution* μ(x)"
* line 209 says "improve ITO learning in three ways" but then four points are listed

---

> ### Author Response · Authors · 2024-11-25
>
> We thank the reviewer for their constructive input. Please find a point-by-point rebuttal below:
> >- Weaknesses:
> >- The paper would benefit from clearer writing and more extensive background sections. It seems to assume that readers are already familiar with the implicit transfer operators framework, which will often not be the case. More explanations and pseudocode (even if it is just in the appendix) would make the paper much easier to read.
> - Response: We agree that the presentation can be improved in the ways the reviewer outlines, in particular adding pseudo-code illustrating the different methods, as it appears to have caused some confusion amongst most reviewers. In the camera ready version we will update the manuscript accordingly.
>
> > - The experiments are conducted only on small-scale systems, and it is unclear how well this approach would scale to more realistic scenarios.
> - Response: While we showed favorable scaling with increasing system size in the original paper, we now also include the application of BoPITO to an all heavy-atom model of Chignolin folding, scaling up by nearly an order of magnitude in system size: https://anonymous.4open.science/r/chignolin-4614/long_term.svg. As for the Prinz and Ala2 examples, we show clear improvements in sample efficiencies in this case as well. Nevertheless, we stress that the limitations of scaling here are primarily caused by the unfavorable scaling of graph neural network architectures. These problems are well known in the community and consequently, there is a lot of ongoing work to help improve the scalability of these methods [1,2].
>
> >- Questions:
> >- The formalism decomposes the operator in a sum of left and right eigenfunctions. Are these eigenfunctions used anywhere in the training objectives directly, or are they solely used to justify/derive the form of the score function ansatz?
> - Response: The eigenfunctions are solely used to justify/derive the form of the score function ansatz. However, we believe using them explicitly is an interesting avenue for future work.
>
> >- In line 139 and 142 it says but also that . How is it possible that Though?
>
> - Response: This is a typo, should be |$\lambda_i|=exp(-k_i)$. For most physical systems, all $\lambda_i$ are positive. We have corrected it now.
>
> > - I am having some trouble understanding the interpolators section. N_max is the maximum training lag, but then we sample with N_int > N_max. Doesn't that cause out-of-distribution problems? Maybe you could add a bit more explanation in the appendix as to why the given model implies an interpolator.
> - Response: Under our setting, this does not cause out of distribution problems because
> 1) we fix N=N_max as the argument of $s_{dyn}$. This makes sure that none of our neural networks are evaluated outside of the parameters observed during training.
> 2) Because we alternate long-lag interpolation steps with short-lag non-interpolation steps for local relaxation. This causes that if our interpolator samples high-energy states (which occurs in practice), these can be relaxed in the short-lag non-interpolation steps. In practice, just one short-lag non-interpolation step works well for us.
> 3) Because we use dynamic observables to fit the interpolation parameter. Without any additional source of information we don’t know how to choose $N_{int}$. However, when we use dynamic observables, we can select the interpolated ensemble which is the most consistent with experimental observables. So this helps us to be more “in-distribution”.
> We added these clarifications to the appendix.
>
> >- line 60: How exactly do you define "off-equilibrium" and "biased"?
>
> - Response: Off equilibrium simulations are unbiased MD simulations whose statistics are not converged, as to faithfully represent the equilibrium statistics. Biased MD are generated using an enhanced sampling technique such as metadynamics or replica exchange. We have added a section in the appendix elaborating on the terminology used in the paper.
>
> >- line 191 and following: what is t_{diff}?
>
> - Response: The diffusion time of the diffusion model. We changed it now in section 2.4 to clarify this.
>
> >Looking at Figure 3, it seems like ITO and BoPITO have very similar performance on short and medium time scales, and BoPITO mostly outperforms ITO on long timescales, presumably due to the enforcement of the Boltzman distribution for long time horizons. I am wondering how often it is precisely the medium time scales that we are interested in though?
>
> - Response: There are cases where the short and medium time-scales are of primary interest to understand molecular mechanisms. In this context, BoPITO does not directly improve sample efficiency. However, the data collection strategies outlined in the paper can also help boost learning medium time-scale dynamics, the BG initialization (Fig 2).
>
> >- Suggestions
> >- Include a section that explains the ITO framework in more detail and includes pseudo-code
> - Response: Added to appendix.

---

> ### Author Response · Authors · 2024-11-25
>
> >- line 107 and other places, I find formulations like "few of the modes of μ(x)" are easier to read if written as "few of the modes of the Boltzmann distribution μ(x)"
>
> - Response: Corrected.
>
> >- line 209 says "improve ITO learning in three ways" but then four points are listed
>
> - Response: Corrected.
>
> [1] The Importance of Being Scalable: Improving the Speed and Accuracy of Neural Network Interatomic Potentials Across Chemical Domains. Eric Qu et al. NeurIPS 2024. https://arxiv.org/abs/2410.24169
>
> [2] Efficient 3D Molecular Generation with Flow Matching and Scale Optimal Transport. Ross Irwin et al. https://arxiv.org/abs/2406.07266

---

> > ### Comment · Reviewer_hu2r · 2024-11-26
> >
> > Thank you for clarifying all the questions and conducting more experiments.
> > However, I think you have not yet uploaded a revised document, which makes it difficult to check the additional experiments and writing. Therefore, I will have to keep my score for now.

---

> ### Author Response · Authors · 2024-11-27
>
> Dear reviewer,
>
> We have updated the manuscript addressing your comments. Please let us know if you have any further questions or concerns.

---

> > ### Comment · Reviewer_hu2r · 2024-11-29
> >
> > Thank you. All my concerns have been addressed, and I will increase my score. Interesting paper!

---

> > > ### Author Response · Authors · 2024-12-03
> > >
> > > Thank you for supporting for our work through your constructive feedback, engagement, and for raising the score!

---

### Official Review · Reviewer_nG3C · 2024-10-28

**Soundness:** 3
**Presentation:** 4
**Contribution:** 3
**Rating:** 8
**Confidence:** 4

**Summary:**

This paper proposes to combine Boltzmann Generators (BG) with Implicit Transfer Operators (ITOs). ITOs are limited because a lot of unbiased molecular dynamics (MD) data is needed for training. This framework combines ITOs with a BG before being able to sample efficiently and be asymptotically unbiased from the equilibrium distribution.

From a more practical standpoint, the score function of the Score-based Diffusion Model is separated into an equilibrium contribution (from the BG prior) and the time-dependent component (to tackle off-equilibrium simulations). BoPITO defines an interpolation between models trained on off-equilibrium simulations and the equilibrium distribution, thus making it a more efficient sampler of long-time MD simulations.

**Strengths:**

- The paper is clearly written and easy to follow. Overall, it represents a pleasant read.
- The authors made a great effort to make the paper self-contained, including a lot of background sections covering the basics of MD, BGs, ITOs, and diffusion models.
- Numerical results are convincing in showing advantages compared to ITOs (PRO), although the approach is not compared to other baselines/methods in the literature (CON). This allows us to only partially appreciate the advantages of the algorithm.
- The authors address the limitations of the current methods and envision possible avenues for future work.
- The code attached as additional material looks good, properly formatted, and seems to run out-of-the-box

**Weaknesses:**

- I find the main weakness of this work being the lack of comparison with other ML-based approaches that can deal with sampling at different time scales (e.g., [3,4]). Moreover, it would also be useful to show a ground truth in the plots when possible, e.g., Fig. 5. Adding more baselines would certainly improve the manuscript.
- As a suggestion, rather than a weakness, I feel it would be better to move the discussion to the related work at the beginning of the paper. This would help people to associate some works they may already know with the present work in case they are familiar with the literature. This would represent a useful prior to have before getting into the core of the paper.
- At the bottom of page 4, the authors say that BoPITO improves ITO learning in three different ways. However, in what follows, there are four points. I also believe it would be helpful to list these contributions visually with bullet points.

**Questions:**

- At the bottom of page 2 the authors write: "As a result, most MD data will be ‘off-equilibrium’, e.g. trajectories exploring one or a few of the modes of $\mu(x)$". Is it correct to interpret 'off-equilibirum' in this case as 'non-ergodic'?
- In line 59-60: " [...] BoPITO, by construction, guarantees asymptotically unbiased equilibrium statistics.". Where do asymptotic guarantees for unbiasedness come from? I presume that reweighting still needs to be enforced in order to have an asymptotically unbiased sampler [1] though it is not immediately clear to me where this occurs.
- in line 116-117: "BGs are trained either with approximately equilibrated simulation data or with biased simulations, from i.e. enhanced sampling, by employing appropriate reweighting[...]". I think this is not entirely correct, strictly speaking. BGs can also be trained in a data-free manner and a combination of data-informed (biased/unbiased) and data-free losses (see [2]). Did the authors solely rely on negative log-like hood training or also on data-free losses (e.g., Reverse KL)?
- I believe it would be useful to have a reference around line 120 about sampling with reweighting to enforce asymptotic unbiasedness, see e.g., [1].
- What do the authors mean by **importance re-sampling**?
- When looking at the results in Fig. 2 I am not sure what is the crystal baseline. Is it some SOTA or how is this obtained?
- As a follow-up, question: are there any other ML-based models that can be used in this case in order to prove the goodness of the proposed method? e.g., some other baselines that are worth comparing here such as naive ITOs, or some sort of BGs able to handle MD at different time steps [3].
- Is there an intuitive explanation for having in the top right corner of Fig. 3 the long time scale difference in correlation to be smaller than medium and short?
- What is an intuitive understanding of $N_{\textrm{int}}$? Is that the resolution in the interpolation process between equilibrium and off-equilibrium distribution? In other words, some sort of annealing parameter?
- Are there any baselines to compare against in Fig.5? See also the weakness stated above in terms of lack of comparison with existing methods. For instance, does FAB [4] allow the computation of the same type of dynamics? If yes, how does it compare to BoPITO? If not, why?
- This is more of a general detour question due to my limited knowledge of MD simulations. In line 475 onwards, the authors write: "These approaches, in general, aim to accelerate molecular simulations akin to BoPITO, yet due to the CG operation, the molecular dynamics (kinetics) will be accelerated, and detailed knowledge of the unbiased dynamics are needed to correct this [...]". I am wondering how this can be possible. When coarse-graining a system and running MD at a coarser level, you inevitably lose information on the degrees of freedom at a smaller scale. I am naively wondering if there are ways to rigorously correct for this and still retain asymptotic guarantees in this case.

- Lastly, I am wondering if the following intuition behind this work is correct: I train a core component of the equilibrium distribution using BGs. Then, I use some biased MD data to refine my backbone model in order to capture rare events, making it an 'ergodic sampler'. The idea is to use the ITOs paradigm to have some smooth interpolation between the equilibrium-off-equilibrium regimes encoded in the model in different ways (BG+biased MD data).


## References
- [1] [Nicoli, Kim A., et al. "Asymptotically unbiased estimation of physical observables with neural samplers." Physical Review E 101.2 (2020): 023304.](https://link.aps.org/accepted/10.1103/PhysRevE.101.023304)
- [2] [Noé, Frank, et al. "Boltzmann generators: Sampling equilibrium states of many-body systems with deep learning." Science 365.6457 (2019): eaaw1147.](https://www.science.org/doi/10.1126/science.aaw1147)
- [3] [Klein, Leon, et al. "Timewarp: Transferable acceleration of molecular dynamics by learning time-coarsened dynamics." Advances in Neural Information Processing Systems 36 (2024).](https://proceedings.neurips.cc/paper_files/paper/2023/file/a598c367280f9054434fdcc227ce4d38-Paper-Conference.pdf)
- [4] [Midgley, Laurence Illing, et al. "Flow annealed importance sampling bootstrap." arXiv preprint arXiv:2208.01893 (2022).](https://arxiv.org/pdf/2208.01893)

---

> ### Author Response · Authors · 2024-11-25
>
> Dear reviewer,
>
> Thank you for your valuable feedback. Please find a point-by-point discussion below.
>
> > - Weaknesses:
>
> > - I find the main weakness of this work being the lack of comparison with other ML-based approaches that can deal with sampling at different time scales (e.g., [3,4]). Moreover, it would also be useful to show a ground truth in the plots when possible, e.g., Fig. 5. Adding more baselines would certainly improve the manuscript.
>
> - Response: We compare against ITO, which is a baseline that does not incorporate explicit knowledge of the equilibrium distribution. Methods related to ITO show comparable performance, yet they are only somewhat directly adaptable to our setting due to highly specialized featurization. Timewarp [3] and Flow annealed importance sampling bootstrap [4] aim to model the Boltzmann distribution, not the transition density.
>
> Timewarp generates candidate samples from the Boltzmann distribution with a generative model trained to generate samples separated by long lags in simulations to reduce the correlation between generated samples. These samples are refined via Metropolis-Hastings (MH), correctly targeting the Boltzmann distribution. Moreover, Timewarp cannot accommodate sampling different lag times, which is fundamental in our approach since we separate learning of long-term dynamics from the rest. Moreover, we discuss in the manuscript that samples generated by BoPITO for long-term dynamics can be metropolised in a manner similar to that of Timewarp.
>
> Flow annealed importance sampling bootstrap [4] is a method for sampling the equilibrium distribution of molecules without access to simulation data, so there is no time dependency in the model. We could train a Boltzmann Generator using this approach and subsequently combine it with BoPITO even without having access to enhanced sampling. Still, we leave this or future work for being a different problem than the one addressed here.
>
> The fact that there are no comparable baselines illustrates the novelty of our approach and, therefore, is a strength and not a weakness of our paper.
> > - As a suggestion, rather than a weakness, I feel it would be better to move the discussion to the related work at the beginning of the paper. This would help people to associate some works they may already know with the present work in case they are familiar with the literature. This would represent a useful prior to have before getting into the core of the paper.
>
> - Response: We thank the reviewer for the suggestion we will reorganize the paper for the camera-ready version. Please understand that such a reorganization will require a substantial amount of rewriting and we prioritize using the discussion period to clarify our main findings.
>
> > - At the bottom of page 4, the authors say that BoPITO improves ITO learning in three different ways. However, in what follows, there are four points. I also believe it would be helpful to list these contributions visually with bullet points.
>
> - Response: Corrected.
>
> >- Questions:
> >- At the bottom of page 2 the authors write: "As a result, most MD data will be ‘off-equilibrium’, e.g. trajectories exploring one or a few of the modes of ". Is it correct to interpret 'off-equilibirum' in this case as 'non-ergodic'?
>
> - Response: The simulation mechanism is ergodic, but only asymptotically (infinite time of simulation). Therefore any finite simulation might have unconverged statistics (transition counts to/from different states). With off-equilibrium, we mean that the underlying statistics of trajectories do not represent well those of equilibrium. One option for this is lack of ergodicity in the trajectories, as the reviewer highlights, but even having explored the full state space is possible to have off-equilibrium trajectories because of rare events, which require to be sampled “many” times to accurately represent equilibrium statistics. We realize that this terminology was unclear in our manuscript and we have updated it accordingly.
>
> > - In line 59-60: " [...] BoPITO, by construction, guarantees asymptotically unbiased equilibrium statistics.". Where do asymptotic guarantees for unbiasedness come from? I presume that reweighting still needs to be enforced in order to have an asymptotically unbiased sampler [1] though it is not immediately clear to me where this occurs.
>
> - Response: Unbiasedness is only guaranteed in the limit of long lags (asymptotic) and it comes from the observation that long-term dynamics is equilibrium, so we can re-weight wrt to the Boltzmann distribution. For long enough lags, it’s done in the same way as in Boltzmann Generators [1]. For a long enough lag, we sample the model evaluating likelihoods and compute Boltzman weights. Assigning these statistical weights to our samples gives an unbiased ensemble. We can also use more elaborate re-weighting methods as in [4].

---

> ### Author Response · Authors · 2024-11-25
>
> > - in line 116-117: "BGs are trained either with approximately equilibrated simulation data or with biased simulations, from i.e. enhanced sampling, by employing appropriate reweighting[...]". I think this is not entirely correct, strictly speaking. BGs can also be trained in a data-free manner and a combination of data-informed (biased/unbiased) and data-free losses (see [2]). Did the authors solely rely on negative log-like hood training or also on data-free losses (e.g., Reverse KL)?
>
> - Response: We train the BGs used in our experiments on data -- here, noise prediction in the DDPM framework. Energy-based training is indeed also an option. However, the computational cost of evaluating divergences renders these 'data-free' losses impractical. Architectural or algorithmic improvements are needed to make that training modality feasible for CNFs and DDPMs.
>
> > - I believe it would be useful to have a reference around line 120 about sampling with reweighting to enforce asymptotic unbiasedness, see e.g., [1].
>
> - Response: Added.
>
> > - What do the authors mean by importance re-sampling?
> - Response: Importance re-sampling is the process of generating an ensemble by resampling samples from a surrogate with probabilities proportional to their importance weights. This terminology was previously used in [5].
>
> > - When looking at the results in Fig. 2 I am not sure what is the crystal baseline. Is it some SOTA or how is this obtained?
>
> - Response: Crystal is an ITO model trained on trajectories initialised at only point in conformational space (x=0.75) and BG is an ITO model trained on trajectories initialised with samples of a Boltzmann Generator.
>
> > - As a follow-up, question: are there any other ML-based models that can be used in this case in order to prove the goodness of the proposed method? e.g., some other baselines that are worth comparing here such as naive ITOs, or some sort of BGs able to handle MD at different time steps [3].
>
> - Response: Unfortunately, there are no comparable baselines available to our best knowledge. See our first response. We are using naive ITOs (not BoPITOs with our proposed factorization) here to isolate the gains from initialising the tractories using a BG (Figure 2) to the gains from the inductive biases (Fig. 3,4 and 5).
>
> > - Is there an intuitive explanation for having in the top right corner of Fig. 3 the long time scale difference in correlation to be smaller than medium and short?
>
> - Response: We consider this a success of our approach. Because of the long-term dynamics inductive bias in BoPITO models, long-term dynamics are well recovered by BoPITO independently of the number of unbiased trajectories. The model needs to learn from the unbiased MD trajectories for both short and medium time scales. Suppose the statistics in the MD trajectories are insufficient to learn them well (more clearly observed in the medium time scales). In that case, BoPITO models can perform better on short and medium time scales than long ones. BoPITO performs better because, in this experiment, we are under a scarce number of unbiased MD trajectories (but not scarce equilibrium data). As more data is added, the gap between ITO and BoPITO closes, as expected. The ITO plot to the left also supports this statement: short and medium time scales are learned at very similar rates in ITO and BoPITO, but BoPITO learns the long-time scale dynamics with much less data.
>
> > What is an intuitive understanding of N_int? Is that the resolution in the interpolation process between equilibrium and off-equilibrium distribution? In other words, some sort of annealing parameter?
>
> - Response: N_int is the number simulation steps under interpolation sampling. However it cannot be interpreted as the number of physical simulation time-steps when doing interpolation because we need to keep the dynamics score model fixed.
>
> > Are there any baselines to compare against in Fig.5? See also the weakness stated above in terms of lack of comparison with existing methods. For instance, does FAB [4] allow the computation of the same type of dynamics? If yes, how does it compare to BoPITO? If not, why?
>
> - Response: To the best of our knowledge BoPITO is the first method enabling interpolation between a model of MD and the equilibrium. We believe this is a strength of our work rather than a weakness. As previously highlighted, FAB [4] models the equilibrium distribution, not the transition density, so it cannot be used as a baseline.

---

> ### Author Response · Authors · 2024-11-25
>
> > - This is more of a general detour question due to my limited knowledge of MD simulations. In line 475 onwards, the authors write: "These approaches, in general, aim to accelerate molecular simulations akin to BoPITO, yet due to the CG operation, the molecular dynamics (kinetics) will be accelerated, and detailed knowledge of the unbiased dynamics are needed to correct this [...]". I am wondering how this can be possible. When coarse-graining a system and running MD at a coarser level, you inevitably lose information on the degrees of freedom at a smaller scale. I am naively wondering if there are ways to rigorously correct for this and still retain asymptotic guarantees in this case.
>
> - Response: We here refer to explicit operator learning approaches, where spectral matching can be used to recover the unbiased time-scales. Practically, this corresponds to knowing the true time-scales, and adjusting the operator models spectrum to conform to these constraints. While this is a conceptually attractive approach, this information is often not available and consequently these methods are not commonly used.
>
> > - Lastly, I am wondering if the following intuition behind this work is correct: I train a core component of the equilibrium distribution using BGs. Then, I use some biased MD data to refine my backbone model in order to capture rare events, making it an 'ergodic sampler'. The idea is to use the ITOs paradigm to have some smooth interpolation between the equilibrium-off-equilibrium regimes encoded in the model in different ways (BG+biased MD data).
>
> - Response: Overall we agree. Just a minor addition: The BG contains only long term dynamics information, but this information is the hardest to learn for ITO (limited statistics, see Figure 3). BoPITO naturally combines the two to combine the strengths of both methods.
>
> References
>
> [1] Nicoli, Kim A., et al. "Asymptotically unbiased estimation of physical observables with neural samplers." Physical Review E 101.2 (2020): 023304.
>
> [2] Noé, Frank, et al. "Boltzmann generators: Sampling equilibrium states of many-body systems with deep learning." Science 365.6457 (2019): eaaw1147.
>
> [3] Klein, Leon, et al. "Timewarp: Transferable acceleration of molecular dynamics by learning time-coarsened dynamics." Advances in Neural Information Processing Systems 36 (2024).
>
> [4] Midgley, Laurence Illing, et al. "Flow annealed importance sampling bootstrap." arXiv preprint arXiv:2208.01893 (2022).
>
> [5] Transferable Boltzmann Generators. Leon Klein, Frank Noé. NeurIPS 2024. https://arxiv.org/abs/2406.14426

---

> > ### Comment · Reviewer_nG3C · 2024-11-26
> > **Acknowledgment for rebuttal**
> >
> > I thank the authors for their detailed and exhaustive reply.
> >
> > All my concerns have been addressed, and I'll raise my score.

---

> > > ### Author Response · Authors · 2024-11-27
> > >
> > > Thank you so much for helping us improve the work through your active engagement and for raising your score!

---

> ### Author Response · Authors · 2024-11-27
>
> Dear reviewer,
>
> We have updated the manuscript addressing your comments. Please let us know if you have any further questions or concerns.

---

### Official Review · Reviewer_4QuU · 2024-11-02

**Soundness:** 3
**Presentation:** 2
**Contribution:** 2
**Rating:** 6
**Confidence:** 3

**Summary:**

The authors point out that a prior work Implicit Transfer Operator (ITO) relies on extensive unbiased simulation, and suggest Boltzmann priors to improve the ability to approximate dynamics. To be specific, the authors first train a score model using the equilibrium data, afterwards fix the score model and only train the time-dependent components with unbiased and possibly off-equilibrium MD data.

**Strengths:**

The paper applies Boltzmann generators in ITO models to capture modes that have very small probability mass in the Boltzmann distributions.

**Weaknesses:**

1. Pre-trained Boltzmann generator (BG)

My main concern with this paper is that it assumes a pre-trained BG. Since BG is trained using the energy function and data samples (more detail in Q2), I am a bit confused about what the authors intended. Is BG only used to generate initial conditions for simulations, for various sampling across the configuration space?

2. Lack of detail of less extensive use of data compared to ITO

One of the important limitations the authors pointed out was the need for extensive,  unbiased molecular dynamics data. However, I do not see any results from the experiment nor analysis that BoPITO succeeds in generating results similar to ITO with less data, especially for Alanine Dipeptide.

3. Limited baselines and molecular systems
    - While the authors claim that ITO can be improved with BG, however, I don’t see enough experimental results to conclude it.
    - Comparison with ITO for molecular systems is done for only Alanine Dipeptide, while ITO has presented qualitative results and quantitative results for the folding process of Chignolin.
    - Comparison of quantitative stationary and dynamic observables to naive molecular dynamics has not been presented, while it has been for ITO for four molecular systems (Chignolin, Trp-Cage, BBA, Villin). To support the claim that BopITO truly predicts time-correlation statistics (dynamic observable), this seems to be needed.

    Overall, it seems that the paper does not present enough experiment results to conclude that BoPITO improves compared to ITO [1].

4. Missing ground truth transition probability for Alanine Dipeptide

One main and important result is the condition transition probability density for Alanine Dipeptide, figure 5. Though the figure shows that BoPITO and ITO (unbiased) show similar results, there are no ground truth data given to compare with. Though I understand it might be impractical to compute the transition densities for alanine dipeptide for 15,000 trajectories of length up to 1 nano-second compared to trajectories of length 500ps as in ITO[1], it would be good if the authors clarified this.

5. Requiring a well-trained Boltzmann generator

Since the proposed method uses Boltzmann generators as priors, a well-trained Boltzmann generator is essential. Although it has rather easily been done for Alanine Dipeptide, it seems that extension to complex molecular systems such as Chignolin is not flexible.

[1] Implicit Transfer Operator Learning: Multiple time-resolution surrogates for Molecular dynamics, NeurIPS 2023

Minor

- Typos
    - Page 4 last paragraph, improve ITO learning in three ways ⇒ fourth ways
- Some notations are not clarified & used
    - $x$: molecular configuration first in page 1, it is the 3D coordinates of molecular system being $x \in \mathbb{R}^{3 \times M}$ where $M$ is the number of particles?
    - Diffeomorphic map $\mathcal{F}_\theta: \mathbb{R}^N \rightarrow \mathbb{R}^N$ , are the dimension of the latent space and configuration space necessarily identical?
- section 2.3 - transfer operators
    - The part explaining equation 5, the eigenvalues, and right and left eigenfunctions sounds awkward.
- Figure 3
    - There are no (a), (b), (c), (d) in the figure, as it is pointed out in section 4.3.
    - The authors ought to show that using BG as priors is better in terms of different time scales and number of trajectories, so it might have been better to plot the columns and legends switched, e.g., ITO / BoPITO and different time scales.
- Figure 5
    - I can’t find the black cross in the first column indicating the initial condition. Could you increase the size of the mark or change the color to something like red?
- Figure 8, 9
    - Explanation on the colors is missing. Could you add a legend for it?
- Equation 15
    - Theta seems to be missing in ${s_{dyn}}$
    - Subscripts for ${x^{t_{diff}}}$ seems to be missing?

**Questions:**

1. Definition regarding bias and off-equilibrium

The papers uses the term biased & unbiased, off-equilibrium. As far as I understood, the definitions could be summarized as the following:
- Off-equilibrium data: simulation results (trajectories) that did not explore some modes of Boltzmann distribution
- Biased simulation & samples: simulation done & sampled achieved by biased molecular dynamics, e.g. enhanced sampling, a modified version of the naive molecular dynamics
- Unbiased simulations & samples: simulation done & sampled achieved by naive molecular dynamics, possibly may not find some modes if the time horizon is too short

To sum up, is the following a correct summarization for the method in section 3.2 and section 4?
- BG are trained with unbiased and equilibrium data for a score model
- Then the time-dependent components score are then trained with unbiased data, possibly off-equilibrium
- Section 4.3 presents that BoPITO can approximate long-term dynamics in the above situation
- Section 4.4 shows that the training data for BG are biased, and BoPITO can recover it?

2. Boltzmann generators in prior

Boltzmann generators [1] were trained by combining two modes: training by energy and training by example.
- Training by energy: Generate a distribution in the configuration space from a Gaussian prior using the model, and minimize the difference between Boltzmann distribution using the energy term as weights.
- Training by example: Since training by energy alone tends to focus on sampling on the most stable meta-stable state, they additionally use training by some valid configurations, e.g., trajectories from naive MD simulations.
Now, my question is about section 4.2 of the paper, where “BG is available before data collection” and “BG trained on equilibrium data” seems to be a contradiction. I may have misunderstood something, could the authors please clarify this part?

3. Derivation of parameterization

In Appendix A.3, the authors derive the score of the transition probability. When deriving equation, is $s_{dyn}(\ldots) = s_{eq}(x^{t_{diff}}, t_{diff}) f_\theta(\ldots) + g_\theta(\ldots)$ ?

4. Comparison to ITO

From Figure 3 to Figure 4, the authors compare ITO and BoPITO. As far as I understood, the difference between these two methods is the parameterized epsilon, i.e. score function. They have separated the equilibrium part and time-dependent components, and have additionally trained the score function by fixing the equilibrium part and only updating the time-dependent term. Are all the other components, i.e., model and data, identical?

[1] Boltzmann generators: Sampling equilibrium states of many-body systems with deep learning, Science 2019

---

> ### Author Response · Authors · 2024-11-25
>
> Dear reviewer, Thank you for your valuable feedback. Please find a point-by-point discussion below.
>
> >Weaknesses:
> >- Pre-trained Boltzmann generator (BG)
> >- My main concern with this paper is that it assumes a pre-trained BG. Since BG is trained using the energy function and data samples (more detail in Q2), I am a bit confused about what the authors intended. Is BG only used to generate initial conditions for simulations, for various sampling across the configuration space?
>
> - Response:  More elaborate point by point answers in Q2.
>
> 1) Boltzmann Generators are surrogate models of the equilibrium density and not necessarily models trained using energy-based learning. We train the BGs used in our experiments using equilibrium data only. We chose the model systems here as extensive simulation data is available, which facilitates designing realistic experiments and strong baselines. Generally, if a BG is unavailable for a system of interest, we could generate data to train one using enhanced sampling. We intend not to diminish the technical challenges associated with BG training; however, we recognize recent strides toward building transferable BGs [1, 2], which significantly broadens the BoPITO approach's scope. We emphasize that the strength of our work is indeed that it allows the combination of the same generative model of off-equilibrium data, generated with unbiased MD and used to train the temporal component of the surrogate, with equilibrium data, potentially generated by enhanced sampling and used for training the Boltzmann Generator.
>
>  2) The BG is not only used for generating diverse initial conditions for simulations, although we show that this helps in Section 3.1. The pre-trained BG is combined with the time-dependent component of the score model to embed a long-term dynamics inductive bias to the model: we make sure that for very long lags for which the system should equilibrate, the surrogate will sample from the Boltzmann Generator/ equilibrium model.
>
> >- Lack of detail of less extensive use of data compared to ITO
> >One of the important limitations the authors pointed out was the need for extensive, unbiased molecular dynamics data. However, I do not see any results from the experiment nor analysis that BoPITO succeeds in generating results similar to ITO with less data, especially for Alanine Dipeptide.
>
> - Response: We apologize for the confusing caption in Figure 3, which is likely the cause of this concern. The metric shown compares each method to statistics from ground truth simulations, so it’s an error measurement. Specifically, we compare how well BoPITO and ITO approximate short, medium, and long time-scale parts of the correlation functions with respect to long-unbiased simulations. We have updated the caption to “Absolute error in predicted time-correlation compared to long unbiased MD simulations,” to clarify this. This Figure shows that BoPITO can recover better long-term dynamics (lower |Δcorrelation|) with less data than ITO.
> Limited baselines and molecular systems
>
> >- While the authors claim that ITO can be improved with BG, however, I don’t see enough experimental results to conclude it.
>
> Response:  Addressed in previous point.
>
> >- Comparison with ITO for molecular systems is done for only Alanine Dipeptide, while ITO has presented qualitative results and quantitative results for the folding process of Chignolin.
>
> - Response: These methods suffer due to their need for extensive training data and scaling for practical utility. In this paper, we address the former of these two problems, which involves data-efficiency, where we show significant improvements even on these small systems and a favorable dependence on system size: e.g., the relative improvement of ALA2 is greater than for the Prinz potential. To strengthen this point, we have extended our experiments to the folding process in an all-heavy atom model of Chignolin (CLN025), where again find substantial improvements in sample efficiencies: https://anonymous.4open.science/r/chignolin-4614/long_term.svg.
>
> >- Comparison of quantitative stationary and dynamic observables to naive molecular dynamics has not been presented, while it has been for ITO for four molecular systems (Chignolin, Trp-Cage, BBA, Villin). To support the claim that BopITO truly predicts time-correlation statistics (dynamic observable), this seems to be needed.
>
> - Response: To address this limitation we go beyond this request show scaling to a much higher dimensional system all heavy atom Chignolin. The results shown for ITO are done for an C-alpha only representation, e.g. 10 particles, whereas the all heavy atom is 93 atoms, an almost 10-fold increase in system size compared to previous work. Consequently, we demonstrate both an improvement in scaling and in data efficiency: https://anonymous.4open.science/r/chignolin-4614/long_term.svg.

---

> ### Author Response · Authors · 2024-11-25
>
> >- Overall, it seems that the paper does not present enough experiment results to conclude that BoPITO improves compared to ITO [1].
>
> - Response: We address sample efficiency gains in point 2 of this rebuttal. The other main advantage of BoPITO is interpolation: ITO models do not allow for interpolation between an equilibrium surrogate and a dynamic surrogate. BoPITO does, and we show that we can approximately recover dynamics of longer lags than observed during training by fitting the interpolator using dynamic observables.
>
> >- Missing ground truth transition probability for Alanine Dipeptide
> >One main and important result is the condition transition probability density for Alanine Dipeptide, figure 5. Though the figure shows that BoPITO and ITO (unbiased) show similar results, there are no ground truth data given to compare with. Though I understand it might be impractical to compute the transition densities for alanine dipeptide for 15,000 trajectories of length up to 1 nano-second compared to trajectories of length 500ps as in ITO[1], it would be good if the authors clarified this.
>
> - Response: Since ITO (unbiased) is trained on abundant data and was already shown to effectively learn the transition density of ALA2 [4] we considered it as ground truth, since it provides higher histogram resolution than a Markov State Model (MSM). However, for clarity we are working on adding the marginal histograms from a MSM baseline as well.
>
> >- Requiring a well-trained Boltzmann generator
> > Since the proposed method uses Boltzmann generators as priors, a well-trained Boltzmann generator is essential. Although it has rather easily been done for Alanine Dipeptide, it seems that extension to complex molecular systems such as Chignolin is not flexible.
>
> - Response: We agree that scaling is one of the current limitations. However, this is not a problem with training Boltzmann Generators per se, but rather a technical problem related to the scaling of computational graphs created for message passing in our neural network model. This is a well-known problem in the community and there are several efforts to design more scalable equivariant architectures [5]. Further, as we have demonstrated above, we can scale BoPITO to a system of the size of Chignolin with current methodology: https://anonymous.4open.science/r/chignolin-4614/long_term.svg.
>
> >- Minor
> >- Typos
> >- Page 4 last paragraph, improve ITO learning in three ways ⇒ fourth ways
>
> - Corrected.
>
> > - Some notations are not clarified & used: molecular configuration first in page 1, it is the 3D coordinates of molecular system being where is the number of particles?
>
> - Thanks, clarified in updated text.
>
> > Diffeomorphic map , are the dimension of the latent space and configuration space necessarily identical?
>
> - Yes.
>
> >- section 2.3 - transfer operators
> > The part explaining equation 5, the eigenvalues, and right and left eigenfunctions sounds awkward.
>
> - We agree the phrasing here was off. We have improved it in the revised version.
>
> >- Figure 3
> >There are no (a), (b), (c), (d) in the figure, as it is pointed out in section 4.3.
>
> - Corrected.
>
> >- The authors ought to show that using BG as priors is better in terms of different time scales and number of trajectories, so it might have been better to plot the columns and legends switched, e.g., ITO / BoPITO and different time scales.
>
> - Reply: We appreciate the feedback. We are considering your suggestion for the final version but are unsure about space constraints because it adds 3 extra subfigures (with Chignolin results) to Figure 3.
>
> >- Figure 5
> > I can’t find the black cross in the first column indicating the initial condition. Could you increase the size of the mark or change the color to something like red?
>
> - Corrected.
>
> > - Figure 8, 9
> > Explanation on the colors is missing. Could you add a legend for it?
>
> - Corrected.
>
> > - Equation 15
> > Theta seems to be missing in
>
> - Corrected.
>
> > - Subscripts for seems to be missing?
>
> - Response: Because N_int does not correspond to physical simulation steps, we prefer to avoid it to reduce possible confusions.
>
> > - Questions:
> > - Definition regarding bias and off-equilibrium
> > The papers uses the term biased & unbiased, off-equilibrium. As far as I understood, the definitions could be summarized as the ?> following:
> > Off-equilibrium data: simulation results (trajectories) that did not explore some modes of Boltzmann distribution
> > Biased simulation & samples: simulation done & sampled achieved by biased molecular dynamics, e.g. enhanced sampling, a > modified version of the naive molecular dynamics
> > Unbiased simulations & samples: simulation done & sampled achieved by naive molecular dynamics, possibly may not find some modes if the time horizon is too short
>
> - Response: That is correct. We have added a section in the appendix clarifying these definitions.

---

> ### Author Response · Authors · 2024-11-25
>
> > - To sum up, is the following a correct summarization for the method in section 3.2 and section 4?
> > BG are trained with unbiased and equilibrium data for a score model
>
> - Response: We train BGs using equilibrium data. This data can be either from very long MD simulations, which we are confident are converged to equilibrium or data generated with enhanced sampling (biased MD). In our experiments, because we have access to long simulations for the systems under study, we train the BGs using long unbiased MD simulation data, but, in general this will not be available. Note, though, that the BG learns to generate independent samples and does not learn time dependencies.
>
> > - Then the time-dependent components score are then trained with unbiased data, possibly off-equilibrium
> - Response: Yes, but now the data is presented to the model in triplets of initial condition, final condition and lag time. This way, the model can learn the time dependent component of the transition density [4].
>
> We would like to add to the summarization of 3.2 that this factorization is novel and that not only allows for improving sample efficiency (especially in the long-term dynamics regime), but also enables interpolation between the mode trained on off equilibrium data and the equilibrium distribution. The interpolation parameter can be chosen to match unbiased correlation function (experimental data) and allows to approximately recover dynamics for longer lag times than those observed during training.
>
> > - Section 4.3 presents that BoPITO can approximate long-term dynamics in the above situation
>
> - Response: Correct, and we show that it gives good estimates even when trained on off-equilibrium data which do not reach global equilibrium.
>
> > - Section 4.4 shows that the training data for BG are biased, and BoPITO can recover it?
>
> - Response: No. This experiment emulates the situation in which the entire conformational space is sampled by, for example, enhanced sampling, but unbiased MD simulations do not sample key transitions between these states due to their low probability. This situation is common. We show that we can overcome this situation by incorporating unbiased dynamic observables, which are experimental data that depend on the hard-to-sample transition.
>
> To simulate this scenario, we remove all transitions associated with the slowest transition in Alanine dipeptide in the unbiased MD simulations; that is, when a transition occurs, we split the trajectories and remove the samples before and after the transition occurs. These data contain biased transition statistics but still represent the equilibrium ensemble well, and we can train a good BG on this data.
>
> We use these data to train a baseline ITO model and show that it never samples the removed transition, as expected. We also train a BoPITO model on these data, but to sample the removed transition, we use BoPITO interpolation instead of sampling normally. We fit the interpolation parameter using the correlation function introduced in eq. 17 and computed using the unbiased MD (the one before removing the transitions). This setting mimics accessibility to experimental data that depends on the hard-to-sample transition [6]. We can accurately recover transition densities for different lags longer than the characteristic time of the removed transition.
>
> > - Boltzmann generators in prior:
> >Boltzmann generators [1] were trained by combining two modes: training by energy and training by example.
> > Training by energy: Generate a distribution in the configuration space from a Gaussian prior using the model, and minimize the >difference between Boltzmann distribution using the energy term as weights.
> >Training by example: Since training by energy alone tends to focus on sampling on the most stable meta-stable state, they >additionally use training by some valid configurations, e.g., trajectories from naive MD simulations. Now, my question is about >section 4.2 of the paper, where “BG is available before data collection” and “BG trained on equilibrium data” seems to be a > contradiction. I may have misunderstood something, could the authors please clarify this part?
>
> - Response: What we are trying to convey is that there may be situations in which a practitioner can trust a transferable generative model of the equilibrium distribution (BG) of the system of their interest e.g. a very similar molecule than one in the training set. In this case, a BG is available before any data collection. Another situation in which this applies is when only enhanced sampling data is available, but not unbiased MD simulations.
>
> >- Derivation of parameterization
> >In Appendix A.3, the authors derive the score of the transition probability. When deriving equation, is ?
> - Response: Yes. We clarified it now.

---

> ### Author Response · Authors · 2024-11-25
>
> > - Comparison to ITO
> > From Figure 3 to Figure 4, the authors compare ITO and BoPITO. As far as I understood, the difference between these two methods is the parameterized epsilon, i.e. score function. They have separated the equilibrium part and time-dependent components, and have additionally trained the score function by fixing the equilibrium part and only updating the time-dependent term. Are all the other components, i.e., model and data, identical?
>
> - Response: Yes. The neural network used in the time dependent component of BoPITO is exactly the same as the score network for ITO.
>
> [1] Boltzmann generators: Sampling equilibrium states of many-body systems with deep learning, Science 2019
>
> [2] Transferable Boltzmann Generators. Leon Klein et al. NeurIPS 2024. https://arxiv.org/abs/2406.14426
>
> [3] Generation of conformational ensembles of small molecules via surrogate model-assisted molecular dynamics. Juan Viguera Diez et al. Machine Learning: Science and Technology. 5:2. DOI: 10.1088/2632-2153/ad3b64. https://iopscience.iop.org/article/10.1088/2632-2153/ad3b64
>
> [4] Implicit Transfer Operator Learning: Multiple Time-Resolution Surrogates for Molecular Dynamics. Mathias Schreiner et al. NeurIPS 2023. https://arxiv.org/abs/2305.18046
>
> [5] The Importance of Being Scalable: Improving the Speed and Accuracy of Neural Network Interatomic Potentials Across Chemical Domains. Eric Qu et al. NeurIPS 2024. https://arxiv.org/abs/2410.24169
>
> [6] Rescuing off-equilibrium simulation data through dynamic experimental data with dynAMMo.  Machine Learning: Science and Technology. 4:4, DOI:  10.1088/2632-2153/ad10ce https://iopscience.iop.org/article/10.1088/2632-2153/ad10ce

---

> ### Author Response · Authors · 2024-11-27
>
> Dear reviewer,
>
> We have updated the manuscript addressing your comments. Please let us know if you have any further questions or concerns.

---

> > ### Comment · Reviewer_4QuU · 2024-11-28
> >
> > I thank the authors for the detailed response and updated manuscript, explaining the use of BG for improving ITO, and additional experiments for a higher dimensional system, i.e., chignolin, clarifying that BoPITO scales up to larger molecule systems.
> >
> > My concerns and questions have mostly been resolved, and I have raised my score accordingly.

---

> > > ### Author Response · Authors · 2024-12-03
> > >
> > > Thank you for helping us improving the manuscript with your exhaustive feedback, and for raising the score!

---

### Official Review · Reviewer_r47p · 2024-11-10

**Soundness:** 2
**Presentation:** 3
**Contribution:** 2
**Rating:** 6
**Confidence:** 4

**Summary:**

This work looks at predicting molecular dynamics simulations quickly over time, using longer time steps. The authors introduce a method using diffusion models to train on both unbiased and biased MD data. The diffusion model is used to estimate the transition density, with the goal of simulating MD more quickly. BoPITO, using Boltzmann priors, is supposed to be more sample efficient than ITO (Implicit Transfer Operator). In BoPITO, the authors also use pre-trained models of the equilibrium distribution to improve the ITO method. The authors test the method on the 1D Prinz potential and alanine dipeptide.

**Strengths:**

- This method could enable running MD simulations over much longer timescales
- By incorporating Boltzmann priors into the training process, this could allow more data efficiency and also ensure recovering the Boltzmann distribution over time

**Weaknesses:**

- There are many ways to do molecular dynamics simulations. A common way is through methods like machine learning potentials. While the time steps that are taken there are smaller, these potentials are much more accurate and have been used for more diverse systems. It would be helpful if the authors provided some comparison to this method, as well as other methods that also try to directly predict molecular dynamics.

- The authors have only run this on two very toy systems: the 1D Prinz potential and alanine dipeptide. These systems are very far off from getting closer towards practical utility of this method. It would be more convincing if there were other systems studied, such as larger proteins or systems like the polymers or electrolytes system from Fu et al. (2023) TMLR.


- This method does not seem to guarantee important properties such as energy conservation.

**Questions:**

- I do not fully follow the advantage of BoPITO recovering the unbiased Boltzmann distribution over long time horizons: isn’t this something that only applies to equilibrium distributions? But at the same time, it seems like the authors are saying they can incorporate off-equilibrium data and/or estimate something about off-equilibrium simulations?

- Can the authors compare their method to other machine learning methods for molecular dynamics simulation?

- Is energy being conserved? How can you guarantee physical dynamics?

- Can the authors show that this method has potential to be broadly applicable by seeing if it works for harder systems, such as larger proteins, or polymers?

---

> ### Author Response · Authors · 2024-11-25
>
> Dear reviewer,
> Thank you for your valuable feedback. Please find a point-by-point discussion below.
>
> > Weaknesses:
> >- There are many ways to do molecular dynamics simulations. A common way is through methods like machine learning potentials. While the time steps that are taken there are smaller, these potentials are much more accurate and have been used for more diverse systems. It would be helpful if the authors provided some comparison to this method, as well as other methods that also try to directly predict molecular dynamics.
>
> - Response: The field of machine-learned interatomic potentials (MLIP) has indeed made dramatic advances in recent years. While this activity is very impressive, it is orthogonal to the work we present in our manuscript. BoPITO aims to address the sampling/simulation problem. MLIPs learn a potential energy function, whereas we learn a transition density, which models the molecular dynamics on a distribution level, e.g., how an ensemble of simulations started from an initial condition evolves over time. We do compare our methods against MD simulation base-lines, however, we stress that these have to be done with the same potential energy model to be meaningful. Still in future work it could be interesting to combine BoPITO with accurate data generated with MLIPs.
>
>
> >- The authors have only run this on two very toy systems: the 1D Prinz potential and alanine dipeptide. These systems are very far off from getting closer towards practical utility of this method. It would be more convincing if there were other systems studied, such as larger proteins or systems like the polymers or electrolytes system from Fu et al. (2023) TMLR.
>
> - Response: We agree that currently, these methods suffer due to their need for extensive training data, and scaling for practical utility. In this paper, we address the former of these two problems, which involves the data-efficiency, where we show significant improvements even on these small systems, and a favourable dependence with system size: e.g. the relative improvement of ALA2 is greater than for the Prinz potential. Now adding data for an all heavy-atom model for the larger system Chignolin (CLN025, 93 atoms or 279 dimensions), we show these results scale favourably with system size: https://anonymous.4open.science/r/chignolin-4614/long_term.svg.
>
> >- This method does not seem to guarantee important properties such as energy conservation.
>
> - Response:  We are unsure of what is meant exactly here:
>
> The goal of our experiments is to learn the transition density of physical systems coupled to a thermal bath and evolving according to stochastic dynamics such as Langevin dynamics. Consequently in this setting, the (total) energy of the subsystem (the molecule in our experiments) is not conserved, it fluctuates stochastically around a constant mean energy. For example, in equilibrium, the probability of observing a conformation, x, follows the Boltzmann distribution, where the probability of observing x depends on its energy. Therefore different energies can have non-zero probability. This is what is meant when people say that the energy is preserved on average (the distribution over long-term dynamics does not change). The total energy of the system (molecule + thermal bath) is indeed conserved, but the thermal bath is not explicitly modeled neither in conventional Molecular Dynamics, nor in our experiments.
> Under deterministic dynamics, it is a natural question to ask if an integration method is energy preserving, i.e. Euler vs Verlet algorithms. There is indeed a large body of literature in “Energy-preserving MD” [1-3]. However, these works use deterministic MD as a proxy for accuracy in stochastic MD. This is not the scope of our work.
>
> Perhaps the reviewer meant if the effective “forces” (the score for small diffusion time [5]) that our models learn resemble a conservative field, i.e. forces are the derivative of some potential energy function. If this is the case, in principle BoPITO can be implemented such that the score model is conservative (e.g. be the derivative of a scalar function); in this context we can treat the BoPITO model as a continuous normalizing flow which satisfies the continuity equation (under some further mild regularity conditions). This corresponds to at the ensemble level satisfying the energy conservation constraints. Empirically, we and many others [4-6] found that the extra overhead at test time associated with a conservative score model does not improve empirical performance significantly and therefore do not use it in this work.
>
> Finally, empirically we show in Figure 11 that our model generates conformations with similar energies to those of SOTA deep generative models.

---

> ### Author Response · Authors · 2024-11-25
>
> >Questions:
> >
> >- I do not fully follow the advantage of BoPITO recovering the unbiased Boltzmann distribution over long time horizons: isn’t this something that only applies to equilibrium distributions? But at the same time, it seems like the authors are saying they can incorporate off-equilibrium data and/or estimate something about off-equilibrium simulations?
>
> - Response: The long time horizon distribution of MD is the equilibrium distribution, which depends on the simulated ensemble, which might be the Boltzmann distribution. In BoPITO we fix the Boltzmann distribution, and only learn the dynamic part of the transition density, which we can learn from short off-equilibrium MD simulations, e.g. simulations which are not globally equilibrated. The advantage of this approach is that we can divide-and-conquer the learning MD, and we demonstrate better data efficiency. With respect to recovering unbiased statistics, the reviewer is right in that this can only be done exactly for the equilibrium distribution. However it has been successfully done for potentially off-equilibrium long-term dynamics in previous work [9] and BoPITO can do it as well.
>
>
> >- Can the authors compare their method to other machine learning methods for molecular dynamics simulation?
>
> - Response:  We compare against ITO which is a baseline that does not incorporate explicit knowledge of the equilibrium distribution. Methods related to ITO show comparable performance yet are not all directly adoptable to our setting due to highly specialised featurizations. Moreover, for a fair comparison with other related methods, other methods should include energy priors as well, which is not trivial and out the scope of this article.  Finally, our comparisons are always gauged against an explicit MD baseline.
>
>
> >- Is energy being conserved? How can you guarantee physical dynamics?
>
> - Response:  We discuss this question in length in our reply to the limitation stated above.
>
>
> > - Can the authors show that this method has potential to be broadly applicable by seeing if it works for harder systems, such as larger proteins, or polymers?
>
> - Response:  In previous work [7], it was shown that this approach works on fast folding proteins, and with improvements in scaling of equivariant GNNs and transformers (e.g. [4,8]) we expect this to help scale to even larger systems. To strengthen this argument, we are attaching results for an all heavy-atom model representation of the protein Chignolin (CLN025) https://anonymous.4open.science/r/chignolin-4614/long_term.svg.
>
> [1] Energy conservation in molecular dynamics simulations of classical systems. Søren Toxvaerd et al. J. Chem. Phys. 136, 224106. DOI: 10.1063/1.4726728.
>
> [2] Energy-conserving molecular dynamics is not energy conserving. Lina Zhang et at. Phys Chem Chem Phys 13;25(35):23467-23476. DOI: 10.1039/d3cp03515h
>
> [3] Energy Conservation as a Measure of Simulation Accuracy. Peter Eastman et al. https://www.biorxiv.org/content/10.1101/083055v1
>
> [4] The Importance of Being Scalable: Improving the Speed and Accuracy of Neural Network Interatomic Potentials Across Chemical Domains. Eric Qu et al. NeurIPS 2024. https://arxiv.org/abs/2410.24169
>
> [5] Two for One: Diffusion Models and Force Fields for Coarse-Grained Molecular Dynamics. Marloes Arts et al. Journal of Chemical Theory and Computation 19:18. DOI: 10.1021/acs.jctc.3c00702. https://pubs.acs.org/doi/full/10.1021/acs.jctc.3c00702
>
> [6] Should EBMs model the energy or the score? Tim Salimans. ICLR 2021 Workshop EBM https://openreview.net/forum?id=9AS-TF2jRNb
>
> [7] Implicit Transfer Operator Learning: Multiple Time-Resolution Surrogates for Molecular Dynamics. Mathias Schreiner et al. NeurIPS 2023. https://arxiv.org/abs/2305.18046
>
> [8] Efficient 3D Molecular Generation with Flow Matching and Scale Optimal Transport. Ross Irwin et al. https://arxiv.org/abs/2406.07266
>
> [9] Timewarp: Transferable Acceleration of Molecular Dynamics by Learning Time-Coarsened Dynamics. Leon Klein et al. NeurIPS 2023. https://arxiv.org/abs/2302.01170

---

> ### Author Response · Authors · 2024-11-27
>
> Dear reviewer,
>
> We have updated the manuscript addressing your comments. Please let us know if you have any further questions or concerns.

---

> > ### Comment · Reviewer_r47p · 2024-12-02
> > **Response to reviewers**
> >
> > Thank you to the authors for your responses. I am still not quite convinced that other ways to do molecular dynamics simulation (such as machine learning potentials) are orthogonal to the work presented here, as machine learning potentials that are run over long timescales can also be used to recover distribution-level quantities. Contextualizing the work better and looking further at large-scale molecular systems would be helpful for future work. I appreciate adding chignolin, but I think there is still more ways to go to show practical utility, especially as other machine learning/molecular dynamics techniques are now able to get towards simulating large-scale molecular systems. However, I do not think the paper should be blocked because of this. I have updated my score.

---

> > > ### Author Response · Authors · 2024-12-03
> > >
> > > Thank for your raising the score and for suggesting interesting future work directions.

---

### Comment · Area_Chair_DWhw · 2024-11-25
**Last day for reviewers to ask questions to the authors!**

Dear reviewers,

Tomorrow (Nov 26) is the last day for asking questions to the authors. With this in mind, please read the rebuttal provided by the authors earlier today, as well as the other reviews. If you have not already done so, please explicitly acknowledge that you have read the rebuttal and reviews, provide your updated view and score _accompanied by a motivation_, and raise any outstanding questions for the authors.

**Timeline**: As a reminder, the review timeline is as follows:
- November 26: Last day for reviewers to ask questions to authors.
- November 27: Last day for authors to respond to reviewers.
- November 28 - December 10: Reviewer and area chair discussion phase.

Thank you again for your hard work,

Your AC

---

### Meta-Review · Area_Chair_DWhw · 2024-12-20

**Metareview:**

As positives for this paper, reviewers have mentioned the importance of the topic of this paper, focusing on enabling temporal coarse grained molecular dynamics. Making use of cheaply obtainable off-equilibrium data was seen as a promising approach to train more scalable models. The proposed method was also lauded for its soundness and being well-founded in statistical principles. Finally, the code that the authors attached was found to be well written and running out of the box. On the initial smaller scale experiments, the results were found to be promising.

As areas for improvement, a common point raised by multiple reviewers was the limited evaluation on the 1D Prinz potential and Alanine dipeptide. Another common point was the request to add comparisons against other baseline methods, with reviewer r47p asking for comparisons against MD simulations with ML potentials, and reviewer hu2r asking for comparison against other ML methods that can sample at different time scales. Furthermore, several reviewers asked for clarifications, and made suggestions to make the writing clearer and more accessible.

The authors have added experiments on Chignolin, which has addressed one of the main concerns raised by the reviewers. The authors have explained why they have omitted comparisons against MD simulations with ML potentials, explaining this is an orthogonal direction and that it is important to compare methods using the same underlying potential energy surface. Furthermore, the authors have explained why it’s hard to compare against other time-lagged ML methods for the point the paper wants to make. The requests for clarifications have been answered well by the authors and they have incorporated suggestions such as inclusion of pseudocode into their revised paper.

During the rebuttal phase 3 out of 4 reviewers have indicated that most of their concerns were addressed in the rebuttal. Reviewer r47p remained of the opinion that comparison against MD simulations with ML potentials and  demonstrations of larger scale applicability beyond the already added chignolin could improve the paper, but does not think this should block acceptance of the paper. Given the explanations of the authors regarding comparison with MD simulations with ML potentials, which seem reasonable to me, and the already added chignolin results, I conclude that the authors have addressed all major concerns raised by the reviewers. Given the unanimous recommendation to accept this paper among reviewers, I therefore recommend to accept this paper.

**Additional Comments On Reviewer Discussion:**

The authors have provided results for additional experiments that show that the method can be scaled to a system like chignolin, taken into account suggestions by the reviewers, such as including pseudo code, and clarified questions, which have been incorporated in the revised manuscript. This has led to multiple raised scores, and a unanimous agreement among reviewers to recommend acceptance of this paper.

---

### Decision · Program_Chairs · 2025-01-22

Accept (Poster)